# Multi-Dimensional Collaborative Optimization Model for Agricultural Water Rights Based on Water Price Reform under Changing Environment

**Linlin Song [1], Hongshu Wang [1,*] and Liang Ding [2]**

1    College of Economics and Management, Northeast Forestry University, No. 26 Hexing Road, Xiangfang District, Harbin 150040, China; songlinlin_nefu@163.com
2    College of Science, Northeast Forestry University, No. 26 Hexing Road, Xiangfang District, Harbin 150040, China; dl@nefu.edu.cn
*    Correspondence: wanghongshu_nefu@163.com

**Abstract:** Agricultural water rights trading has become an effective means to solve the shortage of agricultural water. However, in the context of uncertainty, the key elements of the water rights trading system and their interactions remain unclear. Therefore, this study constructs a multi-dimensional collaborative optimization model for agricultural water rights based on water price reform under a changing environment. The model quantitatively characterizes the synergistic effects of resource, economic, social, environmental, and ecological objectives on initial water rights allocation and trading. At the same time, the model uses a system dynamics model and intuitionistic fuzzy number to reflect the impact of a changing environment and uncertainty of the management system on water rights allocation and trading. The results show that through water rights trading, the overall coordinated development of farms has been improved, and the economic benefits and water resources utilization efficiency have been improved.

**Keywords:** management of agricultural water rights; system dynamics; multi-dimensional evaluation system; multiple uncertainty

## 1. Introduction

This study proposed a multi-dimensional collaborative optimization model for agricultural water rights based on water price reform under a changing environment (MCO-AWR-WPRCE) to weigh the contradictions among multiple objectives of economic benefits, energy consumption, and water use efficiency of agricultural water rights trading. MCO-AWR-WPRCE can effectively manage water rights in irrigated areas to reduce agri-cultural water use and enhance agricultural benefit value while ensuring agricultural production, providing valuable insights and guidance for current and future water resources management.

Population growth, intensive economic activities, and environmental changes have led to a rapid increase in water demand [1]. Agriculture is the industry that consumes the most freshwater, accounting for about 70% of the total global freshwater consumption [2]. In China, more than half of the cultivated land depends on irrigation to ensure cultivation [3]. As agricultural water use continues to increase, better management of water resources is an effective means to address agricultural water stress [4]. Water rights trading is an important way to optimize water resource allocation by using market mechanisms [5]. Water rights trading can promote the optimal allocation of water resources, maintain the relative balance of agricultural ecosystems, and realize the sustainable use of water resources [6]. Therefore, the management of agricultural water rights is one of the important means to realize the sustainable utilization of agricultural water resources.

Reasonable initial allocation of agricultural water rights is a prerequisite for implementing water rights trading and improving water use efficiency [7]. An effective water

rights regime not only lays the foundation for water demand management, but also helps provide a degree of flexibility for water users and other stakeholders to hand over readily available water in the market [8]. However, the initial allocation of agricultural water rights often contains subjectivity and uncertainty, and it is particularly important to objectively determine the initial allocation method of agricultural water rights [9]. The existing initial allocation methods of agricultural water rights mostly use the weighting method, which has distinct regional characteristics and subjective will [10]. The objective construction of the initial water rights evaluation system is an effective way to solve the subjective empowerment problem [11]. Initial water rights allocation is often influenced by multiple dimensions such as resources (water footprint), economy (economic benefits), society (Gini coefficient), environment (greenhouse gas emissions), and ecology (ecological footprint) [12]. Most of the current studies ignore the influence of multi-dimensional factors on the initial water rights allocation scheme, or only consider the response of a few dimensions, which obviously cannot provide effective guidance for water rights allocation in a comprehensive way. The establishment of the initial water rights allocation evaluation index weight system, assigning index weights to the initial water rights allocation, and evaluating the initial water rights in each region can quantitatively characterize the influence of multi-dimensional closely linked indicators on the initial water rights allocation, which has important practical significance for obtaining a reasonable water rights allocation scheme.

Initial agricultural water rights allocation and water rights trading are two important parts of agricultural water rights management, and the water rights trading process is the core of agricultural water rights management [13]. The improvement of water use efficiency can improve the effectiveness and feasibility of water rights trading, thus promoting the sustainable use of water resources. In the process of initial water rights allocation and water rights trading, water intake, water transfer, and other processes will inevitably lead to energy consumption. However, few studies have considered the impact of water use efficiency and energy consumption on the sustainable management of water rights during water rights trading.

Water right transaction is actually the redistribution of water resources. The redistribution of water resources can improve economic benefits, but some people are conservative because they exaggerate their economic benefits, because the increase in economic benefits will do harm to other aspects (such as environment and resource consumption). According to Liptrot and Hussein, redistribution may reduce farmers' income and increase food prices. In addition, the affected farmers themselves may not benefit from the growth of the water-receiving sector. Therefore, the transaction price and reasonable formulation of water rights are particularly important. The trading price of water rights can solve the contradiction between the static nature of initial water rights allocation and the dynamics of social and economic development [14], and play a key role in the smooth realization of water rights trading. A too low trading price not only damages the interests of the transferor of water rights, but also is detrimental to the conservation and protection of water resources. An excessively high trading price may increase the water cost of the water rights transferor and make the transaction difficult to realize [5]. Therefore, determining the trading price of water rights is not only conducive to creating an effective water rights market, but also is beneficial for solving the contradiction of spatial and temporal distribution of water resources and promoting the optimal allocation of water resources.

The management of agricultural water rights is affected by the initial water rights allocation scheme considering multiple dimensions, energy consumption, and water price setting in the process of water rights trading, and it is inevitably affected by multiple factors such as a changing environment and management system, resulting in uncertainty. Therefore, it is very important to reduce the impact of uncertainty on water rights allocation [15]. In terms of the multi-dimensional initial water rights allocation scheme, the driving factors of each dimensional goal change with time and are related to other dimensional goal-driving factors [15]. In the initial water rights allocation process, considering the dynamic and correlation of the driving factors of each dimension goal is helpful to the

accurate adjustment of initial water rights and the effective balance of multi-dimensional goals. The system dynamics (SD) method can solve complex system structure, handle multi-dimensional variables, multiple feedback, and time-varying problems, and is an effective method to solve the above problems [16]. However, there are few studies on coupling the SD method with the multi-dimensional optimization model of initial water rights allocation. This coupling can optimize and predict the multi-dimensional influencing factors sustainably from the perspective of system theory, and more accurately reflect the impact of future environmental changes on multiple dimensions such as resources, economy, society, environment, and ecology. The management system will cause the deviation or insufficiency of data statistics, increase the uncertainty of data, and then affect the water rights trading model, thus affecting the output water rights allocation scheme. The comprehensive consideration of the changing environment and the uncertainty of the management system are of practical significance for the objective and true management of agricultural water rights.

The MCO-AWR-WPRCE model quantifies the synergistic effects of resources, economy, society, environment, and ecology on the initial water rights allocation, and considers the impacts of multiple uncertainties of a changing environment and management system on the model, so as to obtain a reasonable agricultural water rights allocation scheme. The MCO-AWR-WPRCE model is a theoretical model of water rights trading considering initial water rights allocation and water price reform. By managing water rights trading schemes, agricultural water consumption is reduced, thus achieving the purpose of water saving, ensuring the agricultural production demand in water-deficient areas.

## 2. Materials and Methods

This section is divided into five main modules, including (1) data module; (2) evaluation module: SD is used to construct evaluation models for the five dimensions of resources, economy, society, environment, and ecology, in which water footprint index is adopted for resources dimension, economic benefit index is adopted for economic dimension, Gini coefficient index is adopted for social dimension, greenhouse gas emission index is adopted for environmental dimension, and ecological dimension index is adopted for ecological dimension. (3) Optimization module: an agricultural water rights trading model is constructed. MCO-AWR-WPRCE sets the water rights trading mechanism. When the water rights of a certain farm are not enough to meet the irrigation needs of the current year, the farm needs to buy additional water rights from other farms. The aim of this mechanism is to encourage agriculture in the region to implement water-saving measures to improve the efficiency of regional water resources. On the other hand, the surplus water rights can be sold at a reasonable price, converting the surplus water rights into additional economic benefits. The research builds a water rights trading network, which stipulates that water rights trading should be conducted from other regions nearest to the water shortage area, thus realizing the development of trading strategies under distance constraints. If there are no surplus water rights in the nearby region, MCO-AWR-WPRCE's trading strategy will shift to the next region slightly farther away. Through the evaluation module, the initial allocation weight of water rights is output, the trading price of water rights is output through the total water price model, and the initial allocation volume and trading price of water rights are brought into the optimization model. The whole process is carried out under uncertain conditions. (4) Uncertainty module: SD is used to predict the parameters of each dimension of the evaluation system, intuitive fuzzy number is used to quantify the uncertainty of parameters in management, and fuzzy information is more strongly expressed to more accurately express the subjective attitude of decision makers. (5) Model output module: the specific technology roadmap is shown in Figure 1:

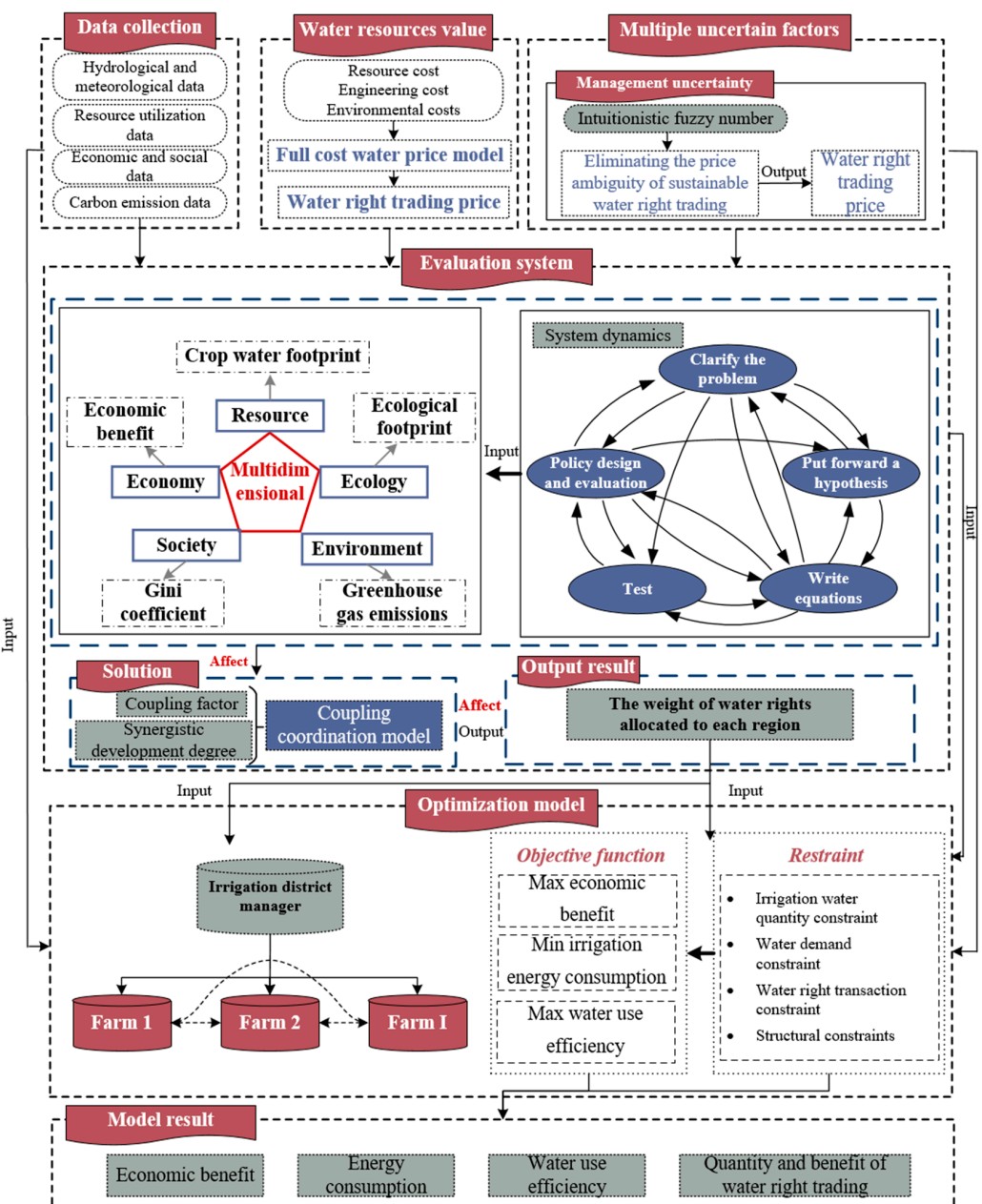

**Figure 1.** Research technology roadmap.

*2.1. Optimization Mode*

2.1.1. Economic Benefit Target Based on Water Rights Trading

In order to better cope with the problem of water scarcity, research is conducted to mitigate the impact of water scarcity by allowing different regions to trade water rights in a decentralized manner [17]. The development of water rights trading can be actively promoted through reasonable initial water rights allocation and water price formulation, which can better manage agricultural water resources and optimize resources utilization efficiency, thus improving the sustainable development of agriculture [18]. Therefore, the water rights trading model is selected as one of the objective functions, and the water rights trading model considering the initial water rights allocation and water price setting is as follows [13]:

$$\max EB^* = \left( \sum_{i=1}^{I} (P_i^* Y_i^* - C_i^*) A_i - WPF_i \cdot WR_i^* A_i - WPB_i \sum_{i=1}^{I} WRB_i A_i + WPS_i WRS_i A_i \right) \tag{1}$$

where $*$ means that the parameter is processed by intuitionistic fuzzy number; $i$ is the farm index, $i = 1 \ldots I$; $EB^*$ is total economic benefit, Yuan; $P_i^*$ is the crop unit price in farm $i$, Yuan/kg; $Y_i^*$ is the crop yield in farm $i$, kg/ha; $A_i$ is the crop area in farm $i$, ha; $WPF_i$ is the current agricultural water price in farm $i$, Yuan/m$^3$; $WR_i^*$ is the initial allocation of water rights in farm $i$, m$^3$/ha; $WPB_i$ is the water purchase price in the water rights trading in farm $i$, Yuan/m$^3$; $WRB_i$ is the amount of water rights purchased by farm $i$ from other farm, m$^3$; $WPS_i$ is the price at which water is sold in the water rights trading in farm $i$, Yuan/m$^3$; $WRS_i$ is the amount of water rights that farm $i$ sells to other farm, m$^3$.

The trading price of water rights, taking into account the transportation cost of water rights, is calculated by the following formula [19]:

$$WPB_i = WPS_i + PE_i \cdot \sum_{i=1}^{I} 1000 \times \left( \sum_{i=1}^{I} \frac{WRB_i h f_{ii} \cdot g}{3.6 \times 10^6} \right) \tag{2}$$

$$h f_{ii} = ph \times TR_{ii} \tag{3}$$

$$ph = 10.294 \times n^2 \times Q^2 \div d^{5.333} \tag{4}$$

where $PE_i$ is the electricity price in farm $i$, Yuan/kWh; $h f_{ii}$ is the head loss along the distance between farm $i$ and other farms, m; $g$ is the acceleration of gravity, N/kg; $ph$ is head loss per unit pipe length, m/m; $TR_{ii}$ is the distance between farm $i$ and other farm $i$, km; $n$ is the roughness rate of the channel in the transportation process, without dimension; $Q$ is channel traffic, m$^3$/s; $d$ is the inner diameter of the channel, m.

When determining the water rights trading, the research takes into account three aspects: resources cost, engineering cost, and environmental cost of water resources [5]. By comprehensively considering resources cost, engineering cost, and environmental cost, the determination of water price in water rights trading can be evaluated and decided more comprehensively. The full-cost water price model can be expressed as follows [20]:

$$WPS_i = WPS_i^R + WPS_i^E + WPS_i^S \tag{5}$$

where $WPS_i^R$ is the cost of water rights resources in farm $i$, including the cost of water resources management and protection, Yuan/m$^3$; $WPS_i^E$ is the engineering cost of water rights in farm $i$, including depreciation cost of fixed assets, maintenance cost of water conservancy facilities, and project operation and management cost, Yuan/m$^3$; $WPS_i^S$ is the environmental cost of water rights in farm $i$, including financial costs, water pollution treatment costs, and water environment compensation costs, Yuan/m$^3$.

Resources cost refers to the quantification of the economic relationship between water resources managers and consumers, reflecting the scarcity value of water resources. The acquisition and treatment of water resources require a certain cost, including the development, collection, transportation, and treatment of water resources. The formula is as follows [20]:

$$WPS_i^E = \frac{Z_i + M_i + S_i}{IQ_i^{mean}} \tag{6}$$

where $Z_i$ is the depreciation cost of fixed assets in farm $i$, Yuan; $M_i$ is the maintenance cost of large equipment in farm $i$, Yuan; $S_i$ is the project operation and management cost in farm $i$, including employee compensation, project maintenance cost, office cost, land use cost, financial cost, and management cost, Yuan.

Engineering cost refers to the quantification of the relationship between production cost and property right, which covers the cost of each link in the process of water resources supply, including the cost consideration of collection, treatment, distribution, and other links. The allocation, transportation, and treatment of water resources need to carry out related engineering construction and operation, which will generate engineering costs. The formula is as follows [20]:

$$WPS_i^S = F_i \cdot WPS_i^R \tag{7}$$

where $F_i$ is the water resources scarcity degree coefficient in farm $i$, without dimension.

Environmental cost refers to the quantification of the compensation degree of water supply agencies to the water environment. The development and utilization of water resources may have an impact on the surrounding ecological environment, so we must pay attention to the impact of water resources on the environment, and take corresponding measures to compensate and protect it. The formula is as follows [20]:

$$F_i = \frac{Q_{nation}}{WR_i} \qquad (8)$$

where $Q_{nation}$ is the national average water consumption for agricultural irrigation, m$^3$/ha.

The distribution of initial water rights takes into account the coordinated development degree of each region. Refer to Section 2.4 for the calculation of the coordinated development degree. The formula for calculating initial water rights is as follows:

$$WR_i^* = D_i^* \cdot IQ_i \qquad (9)$$

where $IQ_i$ is the amount of water rights allocated to farm $i$, m$^3$/ha.

2.1.2. Energy Consumption Targets Considering Water Rights Trading Distance

The process of agricultural irrigation and water rights trading involves energy consumption, and the energy consumption of agricultural irrigation mainly comes from pumping, transportation, irrigation, and other links [21]. This study quantifies the energy consumption in the pumping and conveying process, but does not consider the energy consumption in the irrigation process because of the artesian irrigation method in the study area. However, water rights trading may cause certain energy consumption in other links related to water resources. For example, water rights trading leads to the transfer of water resources, which may require additional pumping equipment (such as pumps) to provide water, thus increasing energy consumption. Therefore, one of the objective functions of the model is to consider the minimum electric energy consumption in the process of irrigation water use and water rights trading, and the calculation formula is as follows [19]:

$$\min P = \sum_{i=1}^{I} \sum_{i=1}^{I} \frac{m_i g h f_{ii}}{3.6 \times 10^6} \qquad (10)$$

$$m_i = 1000 \times \left( WR_i + \sum_{i=1}^{I} WRB_{ii} - WRS_i \right) \times A_i \qquad (11)$$

where $P$ is the energy consumption value of water transmission, kWh; $m_i$ is the quantity of water delivery in farm $i$, kg.

2.1.3. Water Use Efficiency

Water use efficiency can reflect the relationship between soil and water resources utilization and crop yield in agricultural production. It is an important indicator to measure the effect of crops on water resources utilization and a key parameter to evaluate agricultural water management [22]. Improving water use efficiency can not only save water resources, but also reduce energy consumption, reduce environmental pollution, and improve agricultural production efficiency. Therefore, as one of the objective functions of the model, the larger the value of water use efficiency, the better the model. The calculation formula is as follows [22]:

$$\min WUE = \frac{\sum\limits_{i=1}^{I} Y_i^* A_i}{\sum\limits_{i=1}^{I} \left( \left( WR_i + \sum\limits_{i=1}^{I} WRB_{ii} - WRS_i \right) A_i \right)} \qquad (12)$$

### 2.2. Model Constraints

The optimal design of MCO-AWR-WPRCE includes the variable of demand solution (i.e., the decision variable), and the limiting condition of the decision variable is the constraint condition, which can ensure that the model is solved within the feasible boundary. The constraints studied in this chapter include irrigation water quantity, water demand, water rights trading, structure, etc.

(1) Irrigation water constraint

The constraint of irrigation water refers to the restriction and control of the water used in agricultural irrigation to ensure reasonable utilization of water resources and protect the ecological environment. The sum of the initial allocation of water rights for all users should not be greater than the total allocation of water in the irrigation area, thus ensuring water resources security. Constraints can be expressed as follows [5]:

$$\sum_{i=1}^{I} \frac{WR_i A_i}{\eta_{water}} \leq Q^{sur} \tag{13}$$

where $\eta_{water}$ is the water resources availability coefficient, without dimension; $Q^{sur}$ is the available quantity of surface water, m$^3$.

(2) Water demand constraint

Water demand constraint refers to the restriction and control of the water required by the growth and development of crops in each region to ensure the normal growth of crops. The amount of water that can be allocated in each farm needs to meet the water demand of crops to ensure the safe production of food. Constraints can be expressed as follows [5]:

$$W_i^{\min} \leq \left( WR_i + \sum_{i=1}^{I} WRB_{ii} - WRS_i \right) \leq W_i^{\max} \tag{14}$$

where $W_i^{\min}$ is the minimum water demand of crops in farm $i$, m$^3$/ha; $W_i^{\max}$ is the maximum water demand for crops in farm $i$, m$^3$/ha.

(3) Water rights trading constraints

The amount of water rights purchased by each farm from other farms cannot be greater than the amount of water rights sold by other farms, and the amount of water rights sold by each farm cannot be greater than the amount of water rights initially allocated. Constraints can be expressed as follows [5]:

$$0 \leq \sum_{i=1}^{I} WRB_{ii} \leq WRS_i \tag{15}$$

$$0 \leq WRS_i \leq WR_i \tag{16}$$

The model and decision variables should meet the actual situation, that is, the allocation of water rights should not be negative. Constraints can be expressed as follows [5]:

$$WRB_{ii} \geq 0 \quad \forall i \tag{17}$$

$$WRS_i \geq 0 \quad \forall i \tag{18}$$

### 2.3. Model Solution

MCO-AWR-WPRCE is constructed to weigh the three objective functions of economic benefit, energy consumption, and water use efficiency. To couple multiple objectives together, it is necessary to introduce membership function framework [23]. In this paper, a fuzzy algorithm is used to introduce satisfaction variables $\lambda$ to transform the multi-objective

problem into an equivalent fuzzy linear programming problem with a single objective. The transformed model is as follows [23]:

$$\max = \lambda \tag{19}$$

For the objective function that is larger and better, the following membership degree formula is used for quantitative characterization. Maximizing the objective function includes economic benefits and water use efficiency [23]:

$$F(x) - F_{\min}(x) \geq \lambda[F_{\max}(x) - F_{\min}(x)] \tag{20}$$

For the membership function that is smaller and better, the following formula is used for quantitative characterization. The minimization objective function includes the energy consumption objective function [23]:

$$F_{\max}(x) - F(x) \geq \lambda[F_{\max}(x) - F_{\min}(x)] \tag{21}$$

$$\begin{cases} G(x) \leq h \\ x \geq 0 \\ 0 \leq \lambda \leq 1 \end{cases} \tag{22}$$

where $\lambda$ is the satisfaction of the membership function, the larger the $\lambda$ is, the greater the satisfaction of the coordination of each goal and the better the optimization; $F_{\min}(x)$ and $F_{\max}(x)$ are the lowest and highest acceptable levels of the objective function, respectively; $G(x) \leq h$ includes irrigation water constraints, water demand constraints, water rights trading constraints, and structural constraints, which are consistent with the constraints expressed in Section 2.2 above. The model process is shown in Figure 2.

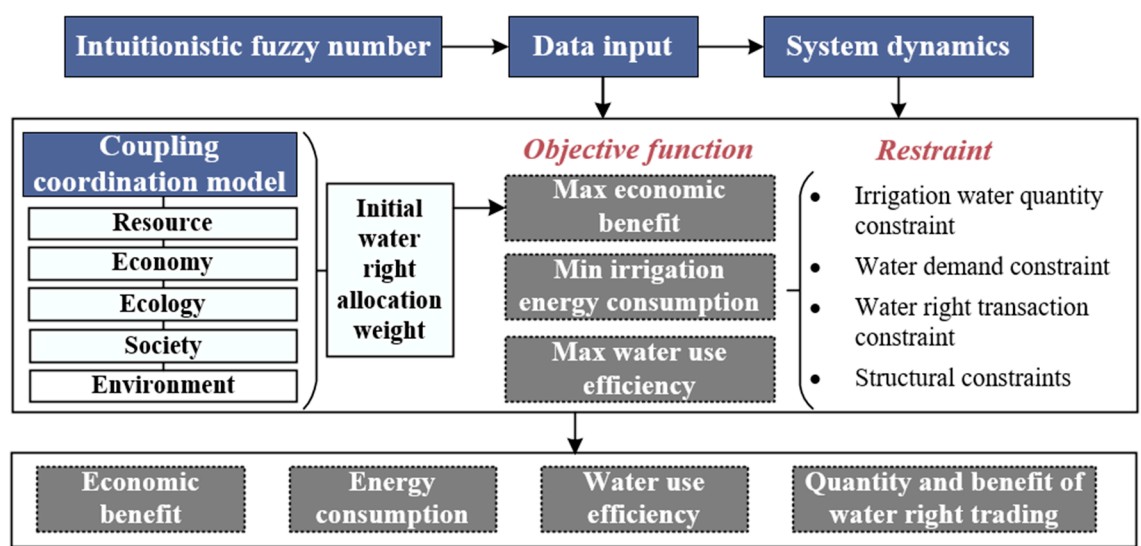

**Figure 2.** Model concept map.

### 2.4. Evaluation Index System for Sustainable Use of Agricultural Water Rights Trading

The evaluation system of agricultural water-saving potential is divided into five subsystems including resources, economy, society, environment, and ecology. This hierarchical structure is helpful to better evaluate the dynamic change characteristics and evolution law of regional agricultural water-saving potential, which will help guide agricultural water use decision making and water-use planning, and promote the sustainable use of regional water resources. The explanations of parameter variables in each dimension are shown in Table 1.

**Table 1.** Definition of rating system parameters.

| Variable Symbol | Variable Interpretation |
| --- | --- |
| $i$ | Farm index |
| $sur$ | Upper corner mark of surface water |
| $gro$ | Upper corner mark of groundwater |
| $*$ | Parameters are fuzzified |
| $RES$ | Resource dimension related parameters |
| $ECO$ | Economic dimension related parameters |
| $SOC$ | Social dimension related parameters |
| $ENV$ | Environmental dimension related parameters |
| $ECOL$ | Ecological dimension related parameters |
| $D_i^*$ | Coordinated development degree in farm $i$, without dimension |
| $C_i$ | Coupling development index in farm $i$, without dimension |
| $T_i$ | Comprehensive development index in farm $i$, without dimension |
| $F_i^{RES}$ | Resources development indicators in farm $i$, m$^3$/kg |
| $F_i^{ECO}$ | Economic development indicators in farm $i$, Yuan |
| $F_i^{SOC}$ | Social development indicators in farm $i$, without dimension |
| $F_i^{ENV}$ | Environmental development indicators in farm $i$, without dimension |
| $F_i^{ECOL}$ | Ecological dimension development index in farm $i$, without dimension |
| $a^{RES}$ | Resources dimension weight, without dimension |
| $a^{ECO}$ | Economic dimension weight, without dimension |
| $a^{SOC}$ | Social dimension weight, without dimension |
| $a^{ENV}$ | Environmental dimension weight, without dimension |
| $a^{ECOL}$ | Ecological dimension weight, without dimension |
| $WFP_i$ | Water footprint in the resources dimension in farm $i$, m$^3$/kg |
| $WFP_i^{blue}$ | Blue water footprint in farm $i$, m$^3$/kg |
| $WFP_i^{green}$ | Green water footprint in farm $i$, m$^3$/kg |
| $WFP_i^{grey}$ | Gray water footprint in farm $i$, m$^3$/kg |
| $WC_i^{blue}$ | Blue water usage in farm $i$, m$^3$ |
| $WC_i^{green}$ | Green water usage in farm $i$, m$^3$ |
| $Yield$ | Total production in farm $i$, kg |
| $A_i$ | Crop planting area in farm $i$, ha |
| $ET_{c,i}$ | Actual evapotranspiration of crops in farm $i$, mm |
| $PR_i$ | Precipitation in farm $i$, mm |
| $Y_i$ | Yield per unit area in farm $i$, kg/ha |
| $WC_i^{grey}$ | Grey water usage in farm $i$, m$^3$ |
| $\theta$ | Nitrogen leaching coefficient, % |
| $FER_i$ | Crop fertilizer usage in farm $i$, kg/ha |
| $c_{nat}$ | Nitrate concentration under normal circumstances, mg/L |
| $c_{max}$ | Maximum nitrate concentration, mg/L |
| $ET_o$ | Reference evapotranspiration of crops, mm |
| $K_s$ | Water stress coefficient of crops, without dimension |
| $K_{cb}$ | Basic crop coefficient of crops, without dimension |
| $K_e$ | Soil evaporation coefficient of crops, without dimension |
| $ECO_i$ | The economic benefits in farm $i$ in economic dimension, Yuan |
| $P_i$ | Crop selling price in farm $i$, Yuan/kg |
| $C_i$ | Crop planting costs in farm $i$, Yuan/ha, including seeds, fertilizers, pesticides, machinery, labor, etc. |
| $CGF_i$ | Crop planting costs in farm $i$, Yuan/ha |
| $CWF_i$ | Crop water cost in farm $i$, Yuan/ha |
| $\delta_i^{fe}$ | Crop fertilizer cost in farm $i$, Yuan/ha |
| $\delta_i^{pe}$ | Crop pesticide cost in farm $i$, Yuan/ha |
| $\delta_i^{ma}$ | Cost of crop machinery diesel in farm $i$, Yuan/ha |
| $\delta_i^{fil}$ | Cost of agricultural film for crops in farm $i$, Yuan/ha |
| $\delta_i^{se}$ | Crop seed cost in farm $i$, Yuan/ha |
| $\delta_i^{lab}$ | Labor cost of crops in farm $i$, Yuan/ha |
| $WPR_i^{sur}$ | Surface water prices in farm $i$, Yuan/m$^3$ |
| $IQ_i^{sur}$ | Surface water irrigation quota in farm $i$, m$^3$/ha |

**Table 1.** *Cont.*

| Variable Symbol | Variable Interpretation |
| --- | --- |
| $WPR_i^{gro}$ | Groundwater prices in farm $i$, Yuan/m$^3$ |
| $IQ_i^{gro}$ | Groundwater irrigation quota in farm $i$, m$^3$/ha |
| $Gini_i$ | Gini coefficient in the social dimension in farm $i$ |
| $PO_i$ | Population in farm $i$, persons |
| $CFP_i$ | Carbon footprint in the environmental dimension in farm $i$, kg CO$_2$-eq/ha |
| $CE_i^{CO_2}$ | CO$_2$ emissions in farm $i$, kg CO$_2$-eq/ha |
| $CE_i^{N_2O}$ | N$_2$O emissions in farm $i$, kg N$_2$O-eq/ha |
| $CE_i^{CH_4}$ | CH$_4$ emissions in farm $i$, kg CH$_4$-eq/ha |
| $SCS$ | CO$_2$ soil carbon sequestration in farm $i$, kg CO$_2$-eq/ha |
| $PES_i$ | Pesticide dosage in farm $i$, kg/ha |
| $DIE_i$ | Agricultural diesel consumption in farm $i$, kg/ha |
| $ELE_i$ | Irrigation electricity consumption in farm $i$, kWh/ha |
| $\varepsilon^{fer}$ | Carbon emission coefficient of fertilizers, kg CO$_2$-eq/kg |
| $\varepsilon^{pes}$ | Carbon emission coefficient of pesticides, kg CO$_2$-eq/kg |
| $\varepsilon^{die}$ | Carbon emission coefficient of agricultural diesel, kg CO$_2$-eq/kg |
| $\varepsilon^{ele}$ | Carbon emission coefficient of irrigation electricity, kg CO$_2$-eq/kg |
| $\varsigma^{CH_4}$ | $CH_4$ emission coefficient of rice field, kg/ha |
| $\gamma^{vol}$ | Nitrogen volatilization coefficient, % |
| $\varsigma^{fer}$ | N$_2$O emission coefficient in fertilizers, kg N$_2$O-eq/kg |
| $\varsigma^{vol}$ | N$_2$O emission coefficient of nitrogen volatilization in fertilizers, kg N$_2$O-eq/kg |
| $\varsigma^{leach}$ | N$_2$O emission coefficient of nitrogen leaching, kg N$_2$O-eq/kg |
| $\varsigma^{SCS}$ | Soil carbon sequestration rate, kg/ha |
| $cef_i^{crop}$ | Ecological footprint of farmland in farm $i$, ha |
| $cef_i^{ene}$ | Ecological footprint of fossil energy in farm $i$, ha |
| $cef_i^{water}$ | Ecological footprint of freshwater resources in farm $i$, ha |
| $EECI_i$ | Ecological coordination index in the ecological dimension, without dimension |
| $CEF_i$ | Crop ecological footprint in farm $i$, ha |
| $CECC_i$ | Ecological carrying capacity of crops in farm $i$, ha |
| $\gamma^{crop}$ | Equivalent factor of farmland, without dimension |
| $Y_{rice}$ | National rice yield per unit area, kg/ha |
| $\gamma^{ene}$ | Equivalent factors of fossil fuels, without dimension |
| $\delta^{FER}$ | Conversion coefficient of fertilizer, without dimension |
| $\delta^{PES}$ | Conversion coefficient of pesticides, without dimension |
| $\delta^{FIL}$ | Conversion coefficient of agricultural film, without dimension |
| $\delta^{DIE}$ | Conversion coefficient of diesel, without dimension |
| $\delta^{ELE}$ | Conversion coefficient of electricity consumption, without dimension |
| $\gamma^{water}$ | Equivalent factor of freshwater resources, without dimension |
| $CV_c$ | Average calorific value of crops, J/T |
| $FO$ | Global freshwater resources output capacity, J/M$^3$ |
| $FD$ | Global freshwater depth, m |
| $\eta$ | Freshwater resources conversion coefficient |
| $j$ | The energy index, with a total energy of $J$ |
| $PA_{ij}$ | Per capita ecological productivity area of energy $j$ in farm $i$, ha |
| $\gamma_{ij}$ | Equivalence factor of energy $j$ in farm $i$, without dimension |
| $y_{ij}$ | Yield factor of energy $j$ in farm $i$, without dimension |
| $\beta$ | Biodiversity coefficient, % |

The agricultural water rights management model covers two key parts, namely initial water rights allocation and water rights trading, between which there is a relationship of mutual influence and restriction [13]. The initial water rights allocation directly affects the amount and frequency of water rights trading that can be conducted. Conversely, an increase in the volume of water rights trading may lead to a decrease in the initial water rights allocation. In order to determine the final allocation of initial water rights, the multi-dimensional agricultural water-saving potential evaluation system constructed in

this study is used to conduct a comprehensive assessment of each region. The evaluation system comprehensively considers many factors and can reasonably allocate initial water rights and ensure the reasonable allocation of resources to meet the agricultural needs of each region.

The degree of correlation of an evaluation system can be measured by the degree of coupling, which is the tightness of multiple connections in the whole [24]. Therefore, the construction of the coordinated development degree model can reflect the level of sustainable coordinated development in various agricultural regions, evaluate the coordination degree of interaction and coupling among various dimensions, and reasonably evaluate the sustainable development level of water rights distribution in various regions [25]. It is composed of coupling degree model and comprehensive development index, and the formula is as follows [25]:

$$D_i = \sqrt{C_i \cdot T_i} \tag{23}$$

The calculated $D_i$ are the data from 2015 to 2020. The intuitionistic fuzzy number is used to reflect the fuzziness, and the accuracy function is used to quantify the definite number and solve it (Section 2.6). Finally, $D_i^*$ is obtained.

The concept of coupling is originally derived from capacity coupling in physics to describe how tightly different parts interact [26]. The coupling degree of this study realizes the dynamic correlation of coordinated development through the interaction and influence of the development indices of five dimensions, namely resources, society, economy, environment, and ecology, and can reflect the degree of interdependence and mutual restriction among the systems. The formula is as follows [26]:

$$C_i = \frac{F_i^{RES} \times F_i^{ECO} \times F_i^{SOC} \times F_i^{ENV} \times F_i^{ECOL}}{\left( \frac{F_i^{RES} + F_i^{ECO} + F_i^{SOC} + F_i^{ENV} + F_i^{ECOL}}{5} \right)^5} \tag{24}$$

Coordination degree refers to the degree of benign coupling in the coupling interaction relationship, which can reflect the quality of coordination [26]. It can be obtained by the weights corresponding to the five dimensions of development index of resources, economy, society, environment, and ecology. The formula is as follows [26]:

$$T = a^{RES} \times F_i^{RES} + a^{ECO} \times F_i^{ECO} + a^{SOC} \times F_i^{SOC} + a^{ENV} \times F_i^{ENV} + a^{ECOL} \times F_i^{ECOL} \tag{25}$$

2.4.1. Resource Dimension Index

Water footprint is an effective index to measure total water resources consumption [27]. Therefore, crop water footprint is included in the evaluation of agricultural water-saving potential rating index system. The crop water footprint family generally includes blue water footprint, green water footprint, and grey water footprint. In the resource dimension, multivariate water footprint is used to quantify the comprehensive consumption of water resources. The formula is as follows [28]:

$$F_i^{RES} = WFP_i = WFP_i^{blue} + WFP_i^{green} + WFP_i^{grey} \tag{26}$$

where $WFP_i^{blue}$ is the blue water footprint of crops, that is, the amount of blue water absorbed from the soil during the growth of crops. Blue water refers to water resources in the surface and soil, usually supplied by deep water sources such as groundwater and rivers [28]. The larger the blue water footprint, the higher the utilization of blue water resources by crops, and the formula is as follows [28]:

$$WFP_i^{blue} = \frac{WC_i^{blue}}{Yield} = \frac{A_i \cdot \max\{0, ET_{c,i} - PR_i\}}{A_i \cdot Y_i} \tag{27}$$

$WFP_i^{green}$ is crop green water footprint, which refers to the water resources stored by natural precipitation in soil and absorbed by crops [28]. The larger the green water

footprint index, the higher the green water resources occupied by crops. The formula is as follows [28]:

$$WFP_i^{green} = \frac{WC_i^{green}}{Yield} = \frac{A_i \cdot \min\{ET_{c\_i}, PR_i\}}{A_i \cdot Y_i} \tag{28}$$

$WFP_i^{grey}$ is the crop grey water footprint, that is, the amount of water required to absorb pollutant load according to water quality standards [28]. The formula is as follows:

$$WFP_i^{grey} = \frac{WC_i^{grey}}{Yield} = \frac{A_i \cdot \theta \cdot FER_i/(c_{\max} - c_{\mathrm{nat}})}{A_i \cdot Y_i} \tag{29}$$

### 2.4.2. Economic Dimension Index

Agricultural economic benefit is the difference between agricultural production labor results and labor consumption, and is a metric to measure the economic distribution of agricultural water rights [29]. Here, labor achievements are mainly quantified by the economic income generated by the sale of crops, while labor consumption is mainly quantified by the economic cost of crops in the planting process. The cost composition involves water, seeds, fertilizers, pesticides, machinery, labor, and electricity consumption, etc. The formula is as follows [29]:

$$F_i^{ECO} = ECO_i = (P_i Y_i - C_i) A_i \tag{30}$$

$$C_i = CGF_i + CWF_i \tag{31}$$

$$CGF_i = A_i(\delta_i^{fe} + \delta_i^{pe} + \delta_i^{ma} + \delta_i^{fil} + \delta_i^{se} + \delta_i^{lab}) \tag{32}$$

$$CWF_i = WPR_i^{sur} \cdot IQ_i^{sur} \cdot A_i + WPR_i^{gro} \cdot IQ_i^{gro} \cdot A_i \tag{33}$$

### 2.4.3. Social Dimension Index

As a function reflecting objective equity, Gini coefficient can effectively measure the difference degree of agricultural water resources allocation between regions [30]. Therefore, the Gini coefficient is chosen as one of the indicators of social dimension. The Gini coefficient values are between 0 and 1, and the smaller the Gini coefficient values, the fairer the distribution is. The formula is as follows:

$$F^{SOC} = Gini_i = \frac{1}{2I \ A_i/PO_i} \sum_{l=1}^{I} \sum_{k=1}^{I} \left| \frac{A_l}{PO_l} - \frac{A_k}{PO_k} \right| \tag{34}$$

### 2.4.4. Environmental Dimension Index

Taking greenhouse gas emissions as an environmental dimension indicator, the aim is to minimize greenhouse gases produced in the process of agricultural production [31]. Therefore, in this study, greenhouse gas emissions are selected as the environmental dimension index for accounting, and the calculation formula is as follows [31]:

$$F_i^{ENV} = CFP_i = \frac{CE_i^{CO_2} + 298CE_i^{N_2O} + 25CE_i^{CH_4} - SCS_i}{A_i \cdot Y_i} \tag{35}$$

where $CO_2$ emissions mainly come from agricultural inputs, including fertilization, pesticide application, agricultural diesel, etc. The calculation formula is as follows [31]:

$$CE_i^{CO_2} = A_i \left( \varepsilon^{fer} \cdot FER_i + \varepsilon^{pes} \cdot PES_i + \varepsilon^{die} \cdot DIE_i + \varepsilon^{ele} \cdot ELE_i \right) \tag{36}$$

$CH_4$ emissions are mainly from rice fields, and the calculation formula is as follows:

$$CE_i^{CH_4} = A_i \cdot \varsigma^{CH_4} \tag{37}$$

N$_2$O emissions are mainly from soil emissions and emissions from fertilizer application, and the calculation formula is as follows:

$$CE_i^{N_2O} = A_i \cdot FER_i \cdot \left( \varsigma^{fer} + \gamma^{vol} \cdot \varsigma^{vol} + \theta \cdot \varsigma^{leach} \right) \tag{38}$$

Carbon fixation can be estimated using empirical coefficients, calculated by the following formula:

$$SCS = A_i \cdot \varsigma^{SCS} \tag{39}$$

2.4.5. Ecological Dimension Index

Agricultural water-saving potential is closely related to ecological benefits, and excessive water saving will lead to regional ecological degradation crisis [32]. The ecological coordination index mainly reflects the coordination degree between regional ecological environment and social and economic development. The closer the ecological coordination index is to 1.414, the better the coordination is. On the contrary, the closer the ecological coordination index value is to 1, the lower the coordination. The ecological index consists of two parts: one is ecological footprint; the other part is ecological carrying capacity [32].

$$F_i^{ECOL} = EECI_i = \frac{CEF_i + CECC}{\sqrt{CEF_i^2 + CECC^2}} \tag{40}$$

The research mainly focuses on the quantification of ecological footprint in the agricultural field. The formula for calculating ecological footprint is as follows [32]:

$$CEF_i = cef_i^{crop} + cef_i^{ene} + cef_i^{water} \tag{41}$$

$$cef_i^{crop} = \gamma^{crop} \frac{A_i \cdot Y_i}{Y_{rice}} \tag{42}$$

$$cef_i^{ene} = \gamma^{ene} \cdot \frac{\left( FER_i \cdot \delta^{FER} + PES_i \cdot \delta^{PES} + FIL_i \cdot \delta^{FIL} + DIE_i \cdot \delta^{DIE} + ELE_i \cdot \delta^{ELE} \right)}{\varepsilon} \cdot cef^{crop} \tag{43}$$

$$cef_i^{water} = \gamma_{water} \frac{A_i \cdot Y_i \cdot CV}{\eta \cdot FO \cdot FD} \tag{44}$$

The formula for ecological carrying capacity is as follows:

$$CECC = (1 - \beta) \times PO \times \sum_{i=1}^{I} \sum_{j=1}^{J} \left( PA_{ij} \times \gamma_{ij} \times y_{ij} \right) \tag{45}$$

*2.5. System Dynamics (SD)*

Future climate change will affect the calculation results of water-saving potential. SD model is used to simulate the driving parameters of agricultural water-saving evaluation system in various dimensions of resources, economy, society, environment, and ecology. The core idea of SD is to study the dynamic connection between various elements within the whole system [33]. Compared to other methods, SD has the outstanding advantage that it can accurately represent the behavior and performance of a system without prior inspection and testing of the system [34]. In the evaluation of agricultural water-saving potential system, the application of SD has shown great potential. By constructing the SD model, the detailed simulation and comprehensive simulation analysis of each key driving factor in the multi-dimensional evaluation system of agricultural water-saving potential can be realized, and then the dynamic evolution process of the system can be fully grasped. Through the operation and adjustment of the model, various potential impacts of agricultural water-saving potential under different future environmental change conditions can be predicted.

SD Simulation Modeling Procedure

To build an SD model, follow these steps to ensure its integrity [34].

1. Identify problem and system boundaries:
   (1) Choose the problem: Clearly define the problem to be solved, including the background and cause of the problem.
   (2) Key variables: Identify the key variables involved, considering the interrelationships between these variables and related concepts.
   (3) Time frame: Clarify the time dimension of the problem, including history, present, and future time periods.
   (4) Reference model: Analyze historical data and behavior of key variables to understand past trends and expected future behavior of the system.

2. Propose dynamic hypothesis:
   (1) Examine existing theories and research to understand how to explain the dynamic behavior of problems.
   (2) Propose the dynamic change hypothesis based on the internal feedback structure of the system.
   (3) Draw diagrams based on initial assumptions, key variables, reference models, and other available data, including system boundary diagrams, subsystem diagrams, causal loop diagrams, stock flow diagrams, policy structure diagrams, etc.

3. Write an equation:
   (1) Master decision rules.
   (2) Specify parameters, behavior relationships, and initial conditions.
   (3) Check if the target is consistent with the boundary.

4. Conduct test:
   (1) Comparison with reference models: Assess whether the model adequately reproduces past behavior patterns.
   (2) Robustness analysis under extreme conditions: In extreme cases, verify if the model's behavior results are consistent with reality.
   (3) Sensitivity analysis: Study how sensitive the model is to parameters, initial conditions, boundaries, and model assumptions.

*2.6. Intuitionistic Fuzzy Number*

Parameter input is required during MCO-AWR-WPRCE construction. Uncertainty and imprecision of input parameters are unavoidable. The constructed MCO-AWR-WPRCE involves many factors such as natural conditions, socio-economic conditions, and human activities, which leads to multi-factor uncertainties in the research process. In the determination of parameters, due to changes in the environment and policies, some parameters lack clear data boundaries, and data information will be omitted if expressed with definite numbers, such as coordinated development degree, crop yield, planting cost, initial water rights allocation, and other parameters change in different times. Intuitionistic fuzzy numbers have more flexible modeling capabilities, stronger interpretability, stronger reasoning ability, and comprehensive integration of uncertainty and fuzzy effects on data [35]. Therefore, intuitionistic fuzzy numbers are used to deal with the uncertainty of the input data of MCO-AWR-WPRCE. However, the intuitionistic fuzzy number method is introduced to transform the time series data into a fuzzy data set, and the mathematical method of precision function can effectively transform the fuzzy data set into the final value.

In 1965, Zadeh proposed fuzzy set theory [36], and then Atannassov proposed the concept of intuitive fuzzy sets [37]. The proposed intuitionistic fuzzy set admits that data have contradictory attributes, which makes the description of data attributes more comprehensive and the performance ability more prominent. In fuzzy decision theory, intuitionistic fuzzy numbers are used to describe the attributes of decision units. In order to

consider the fuzzy multi-attribute decision problem, it is necessary to rank the advantages and disadvantages of the intuitionistic fuzzy numbers, and compare the sizes through the precision function, whose basic formula is referred to in [38].

### 2.7. Sensitivity Analysis and Harmonicity Model

To investigate the impact of various factors on the model's objectives, we conducted sensitivity analysis for each parameter. A sensitivity threshold of |sensitivity| $\geq 0.1$ was employed, where values exceeding this threshold indicate significant influence on the system and close correlation, while values below it suggest minor influence and relative insensitivity. Furthermore, sensitivity analysis reveals both positive and negative sensitivities. A positive value signifies that the variable and parameter change in the same direction, meaning an increase or decrease in the parameter results in a corresponding increase or decrease in the variable. Conversely, a negative value indicates an inverse relationship between the variable and parameter.

$$S = (\Delta x)/(\Delta y) \tag{46}$$

where $S$ is a sensitive result; $(\Delta x)$ is the change value of independent variable (parameter) and $(\Delta y)$ is the change value of dependent variable (multi-objective result).

This study used a two-line average indicator to gain insight into the trade-off between system economic, environmental, and social effects by crop pattern adjustment, and harmony degree was used to express their comprehensive effect on system sustainability, as follows:

$$EES = \frac{1}{2}(Econ \cdot Ene + Ene \cdot Res + Res \cdot Econ)\sin 120° \tag{47}$$

The *EES* metric integrates three main indices of economic (*Econ*)—energy (*Ene*)—resources (*Res*).

### 3. Applications

#### 3.1. Study Area

To verify the feasibility of building MCO-AWR-WPRCE, a real-world case study is used. The case study involves the XKH irrigation area, located in Mishan and Hulin, Heilongjiang province, with geographical coordinates of 132°45′–133°17′ east longitude and 45°01′−45°41′ north latitude. The irrigation area is adjacent to Songacha River and Wusuli River, north to BWL farm Tongsan Highway and Dalian Baohe River, west to Xiaohei River. There are five farms in the region: BWL farm, BWQ farm, BWB farm, QF farm, and XKH farm. The total area is $2.36 \times 10^4$ ha, with $2.11 \times 10^4$ ha of arable land ($0.63 \times 10^4$ ha of paddy field and $1.48 \times 10^4$ ha of dry field) currently available. The five farms under study are state-owned farms, which have management areas, operation stations, and forest farms. Now it has developed into an all-round development of agriculture, forestry, animal husbandry, sideline, and fishery; the national important commodity grain base, which is integrated with science, education, culture, health, and sports, has gradually developed into a specialized farm with rice cultivation as the leading industry. The irrigated area has a continental monsoon climate in the middle and temperate zone, with a large variation in the range of cold and hot air and frequent invasion by cold air. The average annual precipitation is 565 mm, the reference evapotranspiration is 1240 mm, the precipitation distribution is extremely uneven, and the average annual temperature is 2.8 °C, as shown in Figure 3. Five farms have perfect irrigation channels and water diversion and drainage facilities, but the extensive management of water resources leads to the waste of water resources, so XKH irrigation area is selected for water resources management. The above information was obtained through field investigation.

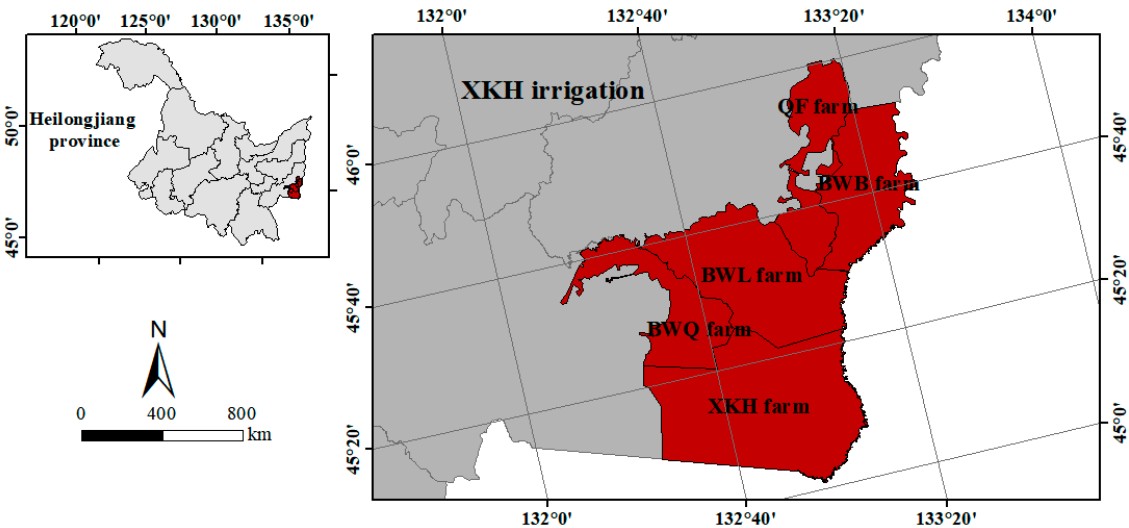

**Figure 3.** Overview of the XKH irrigation.

*3.2. Model Parameters and Data*

MCO-AWR-WPRCE data mainly include the following categories: socio-economic-related parameters, energy-related data, and hydrometeorological-related data. Data sources include China Geospatial Data Cloud, China Earth System Science Data Sharing Network, China Meteorological Science Data Sharing Service network, feasibility study report of XKH irrigation area, planning report, and field investigation.

3.2.1. MCO-AWR-WPRCE-Related Data

The MCO-AWR-WPRCE input includes data on economic targets, energy consumption, and water use efficiency, as shown in Tables 2 and 3. The planting cost of each farm in the XKH irrigation area is 7716.89 Yuan/ha, the transportation price is 0.48 Yuan/m$^3$, and the current water price is 0.003 Yuan/m$^3$. The maximum and minimum constraints for irrigation water are 4500 m$^3$/ha and 2500 m$^3$/ha, respectively. The coordination degree is obtained by solving the model established in Section 2.3, and the initial allocation of water rights in various regions can be determined by the coordination development degree. The above data are all from the field investigation and feasibility study report of the XKH irrigation area.

**Table 2.** Economy-related parameters.

| Name | Crop Price | Engineering Water Prices | Environmental Cost | Selling Water Price |
| Unit | Yuan/kg | Yuan/m$^3$ | Yuan/m$^3$ | Yuan/m$^3$ |
| --- | --- | --- | --- | --- |
| BWL farm | 4.36 | 0.077 | 0.013 | 0.093 |
| BWQ farm | 4.57 | 0.039 | 0.016 | 0.057 |
| BWB farm | 5.37 | 0.043 | 0.012 | 0.059 |
| QF farm | 4.49 | 0.035 | 0.012 | 0.050 |
| XKH farm | 3.71 | 0.048 | 0.012 | 0.062 |

**Table 3.** Society-related parameters.

| Name | Area | Water Rights Allocation | Yield | Coordinated Development Degree | Initial Water Rights |
| Unit | ha | m$^3$/ha | kg/ha | Di | m$^3$/ha |
| --- | --- | --- | --- | --- | --- |
| BWL farm | 57,453 | 3001 | 9585 | 0.80 | 2411.90 |
| BWQ farm | 28,927 | 2580 | 9196 | 1.52 | 3924.06 |
| BWB farm | 32,627 | 3247 | 8713 | 0.44 | 1443.82 |
| QF farm | 26,600 | 3451 | 8751 | 0.99 | 3431.69 |
| XKH farm | 35,813 | 3451 | 8817 | 1.24 | 4266.96 |

### 3.2.2. Rating Model-Related Data

The weight values of $a^{RES}$, $a^{ECO}$, $a^{SOC}$, $a^{ENV}$, and $a^{ECOL}$ in each dimension of the model are 0.2. The SD model is used to simulate the driving parameters of the agricultural water-saving evaluation system in the dimensions of resources, economy, society, environment, and ecology. The above data are all from the field investigation and feasibility study report of the XKH irrigation area. Social dimension parameters are simulated by data from 2011 to 2019, and other dimension data are simulated by data from 2015 to 2019, as shown in Tables 4–7.

**Table 4.** SD model simulation parameters—society dimension.

|  | Time | 2011 | 2012 | 2013 | 2014 | 2015 | 2016 | 2017 | 2018 | 2019 | Unit |
|---|---|---|---|---|---|---|---|---|---|---|---|
| BWL farm | Actual evapotranspiration | 632 | 615 | 633 | 636 | 699 | 647 | 718 | 745 | 790 | mm |
|  | Precipitation | 573 | 603 | 573 | 464 | 584 | 865 | 642 | 653 | 985 | mm |
|  | Fertilizing amount | 227 | 275 | 250 | 227 | 206 | 139 | 138 | 146 | 168 | kg/ha |
| BWQ farm | Actual evapotranspiration | 632 | 615 | 633 | 636 | 699 | 647 | 718 | 745 | 790 | mm |
|  | Precipitation | 573 | 603 | 573 | 464 | 584 | 865 | 642 | 653 | 985 | mm |
|  | Fertilizing amount | 228 | 276 | 251 | 228 | 207 | 182 | 234 | 253 | 135 | kg/ha |
| BWB farm | Actual evapotranspiration | 632 | 615 | 633 | 636 | 699 | 647 | 718 | 745 | 790 | mm |
|  | Precipitation | 573 | 603 | 573 | 464 | 584 | 865 | 642 | 653 | 985 | mm |
|  | Fertilizing amount | 166 | 200 | 182 | 166 | 151 | 164 | 202 | 151 | 134 | kg/ha |
| QF farm | Actual evapotranspiration | 632 | 615 | 633 | 636 | 699 | 647 | 718 | 745 | 790 | mm |
|  | Precipitation | 573 | 603 | 573 | 464 | 584 | 865 | 642 | 653 | 985 | mm |
|  | Fertilizing amount | 187 | 226 | 206 | 187 | 170 | 121 | 116 | 121 | 136 | kg/ha |
| XKH farm | Actual evapotranspiration | 632 | 615 | 633 | 636 | 699 | 647 | 718 | 745 | 790 | mm |
|  | Precipitation | 573 | 603 | 573 | 464 | 584 | 865 | 642 | 653 | 985 | mm |
|  | Fertilizing amount | 141 | 171 | 155 | 141 | 128 | 492 | 492 | 257 | 372 | kg/ha |

**Table 5.** SD model simulation parameters—economic dimension.

|  | BWL Farm | | BWQ Farm | | BWB Farm | | QF Farm | | XKH Farm | |
|---|---|---|---|---|---|---|---|---|---|---|
|  | Yield (kg/ha) | Crop Price (Yuan/kg) | Yield (kg/ha) | Crop Price (Yuan/kg) | Yield (kg/ha) | Crop Price (Yuan/kg) | Yield (kg/ha) | Crop Price (Yuan/kg) | Yield (kg/ha) | Crop Price (Yuan/kg) |
| 2015 | 9375 | 3.55 | 9195 | 3.72 | 8895 | 4.36 | 8700 | 3.59 | 9060 | 2.77 |
| 2016 | 9200 | 4.93 | 9179 | 5.15 | 8881 | 6.04 | 8850 | 4.97 | 8700 | 3.84 |
| 2017 | 9304 | 2.54 | 9742 | 2.73 | 9022 | 3.3 | 9150 | 2.6 | 9405 | 3.38 |
| 2018 | 12366 | 3.45 | 9138 | 3.68 | 7966 | 4.26 | 8669 | 3.87 | 7580 | 3.13 |
| 2019 | 8934 | 6.44 | 9136 | 6.73 | 8632 | 7.89 | 8668 | 6.5 | 9191 | 5.02 |
| 2020 | 8957 | 5.62 | 8868 | 5.88 | 8708 | 6.89 | 8640 | 5.68 | 8775 | 4.39 |

**Table 6.** SD model simulation parameters—environmental dimension.

|  | BWL Farm | | BWQ Farm | | BWB Farm | | QF Farm | | XKH Farm | |
|---|---|---|---|---|---|---|---|---|---|---|
|  | Pesticide Dosage (kg/ha) | Diesel Oil Consumption (kg/ha) | Pesticide Dosage (kg/ha) | Diesel Oil Consumption (kg/ha) | Pesticide Dosage (kg/ha) | Diesel Oil Consumption (kg/ha) | Pesticide Dosage (kg/ha) | Diesel Oil Consumption (kg/ha) | Pesticide Dosage (kg/ha) | Diesel Oil Consumption (kg/ha) |
| 2015 | 4.08 | 0.12 | 2.36 | 0.12 | 8.69 | 0.1 | 4.68 | 0.08 | 1.74 | 0.09 |
| 2016 | 3.87 | 0.16 | 2.34 | 0.13 | 8.44 | 0.1 | 5.85 | 0.07 | 1.73 | 0.09 |
| 2017 | 3.75 | 0.16 | 2.28 | 0.13 | 8.44 | 0.1 | 5 | 0.07 | 1.73 | 0.09 |
| 2018 | 3.83 | 0.15 | 2.22 | 0.12 | 8.44 | 0.1 | 5.78 | 0.06 | 1.73 | 0.09 |
| 2019 | 4.54 | 0.14 | 7.15 | 0.12 | 9.91 | 0.11 | 5.99 | 0.07 | 3.41 | 0.09 |
| 2020 | 4.35 | 0.14 | 7.07 | 0.1 | 9.32 | 0.11 | 5.94 | 0.08 | 3.43 | 0.09 |

After establishing the system dynamics model, it is crucial to assess its validity. This entails testing the model's accuracy by comparing historical data with simulation results to determine their consistency and gauging the reliability of the model. Subsequent to the system dynamics model test, evaluating the model is essential to ensure the precision of its parameter calculations [39,40]. The root mean square error (RMSE) method is employed for this evaluation.

**Table 7.** SD model simulation parameters—social dimension.

|  | BWL Farm | BWQ Farm | BWB Farm | QF Farm | XKH Farm |
|---|---|---|---|---|---|
| **Total Population (Person)** | | | | | |
| 2015 | 20,497 | 17,113 | 12,707 | 13,483 | 9702 |
| 2016 | 20,931 | 17,280 | 13,585 | 13,584 | 12,814 |
| 2017 | 20,765 | 17,210 | 14,332 | 13,670 | 12,736 |
| 2018 | 21,881 | 17,092 | 14,930 | 13,708 | 12,736 |
| 2019 | 22,722 | 17,337 | 15,362 | 13,643 | 9017 |

## 4. Results

### 4.1. SD Prediction Result

The evaluation system of agricultural water-saving potential is a comprehensive multi-dimensional system, including five sub-systems of resources, economy, society, environment, and ecology. In order to better evaluate the water-saving potential of each region, the evaluation system consists of 69 variables, which interact with each other and jointly affect the calculation of agricultural water-saving potential, as shown in Figure 4. Among them, the effective irrigated area, total population, grain output value, carbon emission, and ecological footprint are state variables. Population increase, population decrease, carbon emission increase, carbon emission reduction, biological footprint increase, and farmland area change are rate variables. Fertilizer consumption and grain yield are auxiliary variables. The fertilizer conversion coefficient and land balance factor are constants. It can be clearly seen from the figure that there are intricate internal connections among various dimensions. For example, the total population can be quantified not only by the value of the Gini coefficient, but also by the calculation of the ecological footprint.

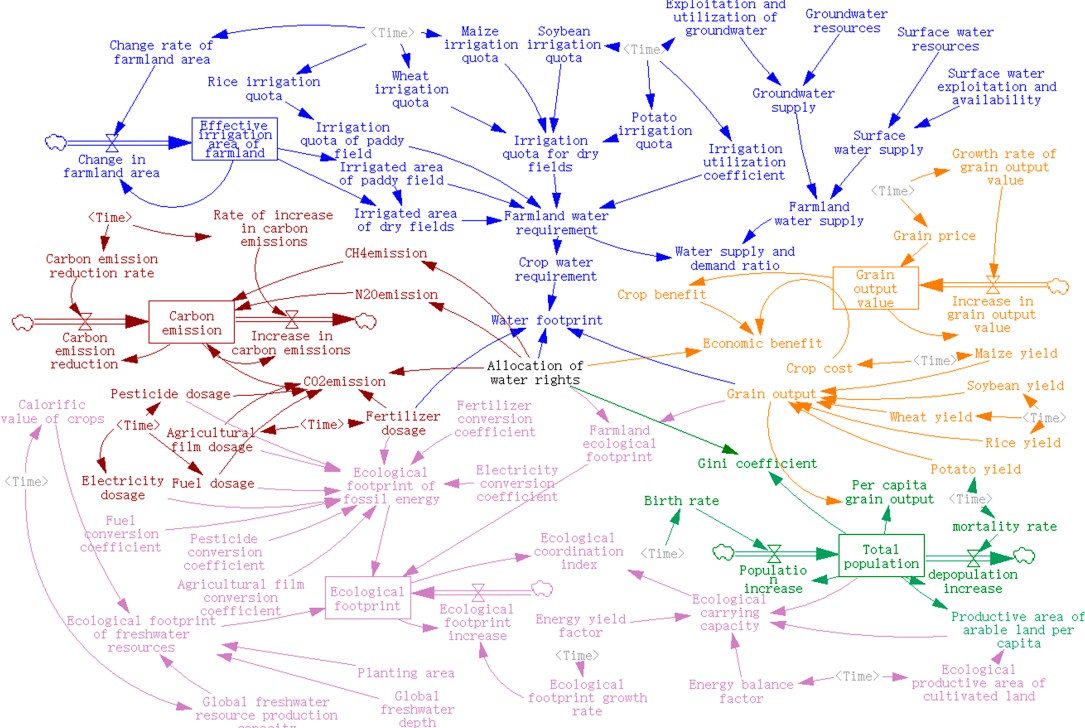

**Figure 4.** Multi-dimensional evaluation of SD flow charts. Note: Blue represents the environmental dimension, red represents the resource dimension, pink represents the economic dimension, green represents the social dimension, and yellow represents the ecological dimension, Gray is the starting point of each dimension and has no practical significance.

In this study, the SD model is used to model and predict each parameter variable in the multi-dimensional agricultural water-saving potential evaluation system. In the resource dimension, the SD model chooses 2011 as the starting time of model operation, and 2011–2019 as the time of the model operation and actual test. In the dimensions of economy, environment, and society, the SD model chooses 2015 as the starting time of the model operation, and 2015–2020 is used to test the correctness of the model, as shown in Figure 5. Since the parameters of the ecological dimension do not have long historical series values, the current annual data are used to calculate in the model.

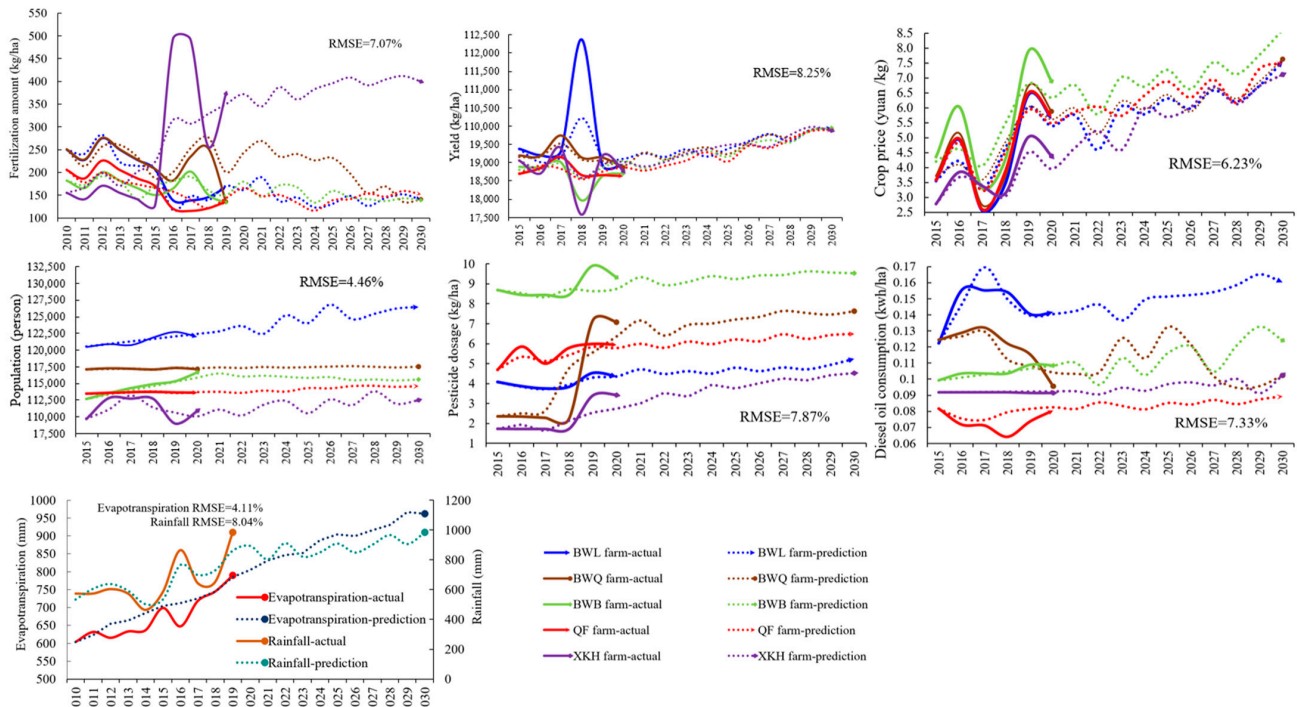

**Figure 5.** Water allocation map after model optimization. Note: Crop growth period 1–5 refers to the dividing stage, jointing stage, heading stage, milk-ripening stage, and yellow-ripening stage of rice.

The error of the simulation results is controlled within 10%, which verifies the reliability of the model. In the resource dimension, key indicators such as actual evapotranspiration, precipitation, and fertilizer application are selected for simulation prediction. Compared with the historical values, it can be seen that the actual evapotranspiration, precipitation, and fertilizer application showed a gradual upward trend, among which the change trend of actual evapotranspiration is relatively gentle, and the change trend relatively between the historical minimum actual evapotranspiration and the predicted maximum actual evapotranspiration is 56.81%. The variation frequency between the historical minimum precipitation and the predicted maximum precipitation is 70.25%. In the economic dimension, the output and crop price are selected to simulate the forecast, and the output and crop price show a gradually rising trend. Yields vary from 6.67% to 14.94%, with different farms varying in frequency. The change trend of the crop price shows a wave growth, with a range of 83.68% to 152.39%. In the social dimension, the index of the population number is selected for the simulation prediction. The variation range of the population number of each farm is between 11.29% and 46.36%, indicating that the change frequency of the population flow is also different in different farms due to their different development. This difference is closely related to the economic development of each farm. Farms with higher economic benefits have greater demand for personnel and greater adhesion to talents. Conversely, farms with lower economic benefits have lower adhesion to talents. In the environmental dimension, pesticide consumption and diesel oil consumption are simulated. The variation range of pesticide consumption is 9.29% to

239.42%, and that of diesel oil is 10.16% to 34.74%. The amount of pesticide and diesel oil shows a gradually increasing trend, and there are obvious differences among different farms. The change trend of pesticide consumption first increases and then decreases, and finally tends to a stable growth trend.

### 4.2. Resource, Economic, Environmental, Social, and Ecological Dimensions Evaluation System Results Analysis

Coordinated development degree D is a standard for evaluating the level of coordinated development among five dimensions of resources, economy, society, environment, and ecology. The obtained coordinated development degree of the evaluation system is analyzed according to the criteria in Table 8.

**Table 8.** Degree of coordinated development D.

| Coordinated Development Level | 0.9–1.00 | 0.8–0.9 | 0.7–0.8 | 0.6–0.7 | 0.5–0.6 | 0.4–0.5 | 0–0.4 |
|---|---|---|---|---|---|---|---|
| Coordination level | High-quality coordination | Good coordination | Intermediate coordination | Primary coordination | Barely coordination | Borderline disorder | Imbalance |

According to the constructed evaluation system, the coordinated development degree of each dimension index in different periods can be calculated, as shown in Figure 6. By analyzing the data results from 2015 to 2020, the average coordinated development degree of each farm and the average growth proportion of each year can be obtained. During this period, the average coordinated development degrees of the five farms are 0.25, 0.47, 0.15, 0.31, and 0.39, respectively. It can be seen that the development status of different farms is quite different. Among them, the BWQ farm has maintained a high level of coordinated development degree, and the BWB farm, although the annual average is the lowest, continues to rise, and its growth rate ranks first among the five farms. Among them, the coordinated development degree of the XKH farm is relatively high and in a positive growth state, with a growth rate of 5.98%. In contrast, the coordinated development degrees of the BWB or BWL farms are relatively low, but their growth rates are high, and they are in a state of positive development. Overall, all five farms are in a state of uncoordination or near uncoordination. To this end, farm managers need to always adhere to the scientific development concept, attach importance to and give play to the unique advantages of each farm in different leading industries and resources fields, and optimize the layout on this basis to achieve the coordinated improvement of economic, social, and environmental benefits. In the process of promoting sustainable agricultural development, factors such as population change and ecological environmental protection should also be taken into account to formulate more scientific and rational development plans and policies.

### 4.3. Analysis of Water Rights Trading Scheme

MCO-AWR-WPRCE considers the influence of the trading distance of water rights on the allocation of water rights, and realizes the efficient allocation of water resources. By minimizing trading distance constraints, the model achieves efficient water rights allocation and energy consumption savings. MCO-AWR-WPRCE not only helps save energy consumption, but also helps improve the economic benefits of agricultural water rights. As shown in Figure 6, through model optimization, it is found that the use of water rights in the BWL and the BWB farm is in a relatively tight state, and water rights need to be purchased from other farms to meet their agricultural development needs. The amount of water rights purchased by the BWL farm is $5.06 \times 10^6$ m$^3$, and the amount of water rights purchased by the BWB farm is $2.48 \times 10^7$ m$^3$ and $9.98 \times 10^6$ m$^3$, respectively, from the QF farm and the XKH farm, in Figure 7. However, the BWL farm, the QF farm and the XKH farm do not carry out water rights trading because the initial water rights provided by them are greater than the water rights demanded under the boundary conditions of the model.

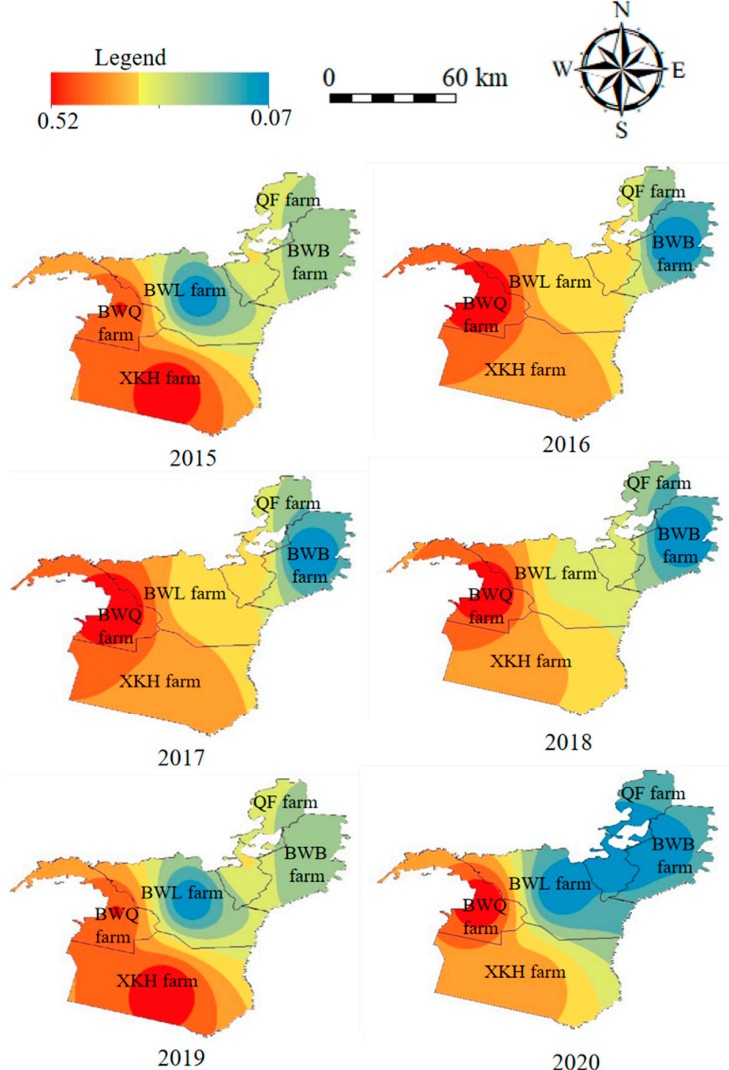

**Figure 6.** Calculation results of coordinated development degree of each farm from 2015 to 2020.

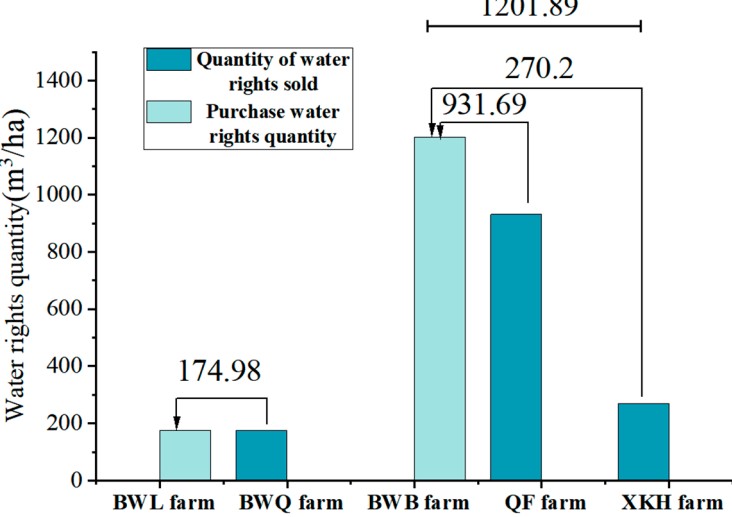

**Figure 7.** Water rights allocation schemes for individual farms.

*4.4. Objective Function Result Analysis*

Based on the water rights trading scheme, this study studied the optimal tradeoff model between multiple objectives such as economic benefit, energy consumption, and water use efficiency, and explored the impact of water rights trading on water resources management and agricultural production. In this chapter, the direct fuzzy number processing method is used to solve each target, and a series of data about each farm is obtained, as shown in Figure 8. Through comparative analysis, it is found that the total economic benefit of the BWL farm is the highest, which is $1.95 \times 10^9$ Yuan, while that of the QF farm is the lowest, which is $8.42 \times 10^8$ Yuan; the economic benefit per unit area of the BWQ farm is the highest, which is $3.82 \times 10^4$ Yuan/ha, and the lowest is the Ba Wuba farm, at only $2.50 \times 10^4$ Yuan/ha. The reasons for these differences are mainly the differences in the planting area and the degree of coordinated development of each farm. The BWQ farm has the highest total energy consumption and the lowest energy consumption per unit area, while the BWB farm has the lowest total energy consumption and the highest energy consumption per unit area. This is mainly because energy consumption considers the initial allocation of water rights and the energy allocation of water rights trading. The highest water use efficiency is 1.95 kg/m$^3$ in the BWL farm, and the lowest is in the BWB farm, at only 1.84 kg/m$^3$. The differences in each indicator confirm the differences in the management of water rights between farms. On this basis, this study further discusses the impact of water rights trading on water resources management and agricultural production in irrigated areas. Through comparison, it is found that the total economic benefit of each farm increased by 2.25% and the water interest efficiency increased by 7.43% compared with the actual situation. This indicates that water rights trading can reduce agricultural water use and increase agricultural benefit value on the premise of ensuring agricultural production. Therefore, the introduction of a water rights trading model in the management of water rights in irrigated areas can provide valuable insights for current and future water resources management. Water rights trading can reduce agricultural water use and increase agricultural benefit value on the premise of ensuring agricultural production. This is of great significance to the practice of water right management in irrigated areas.

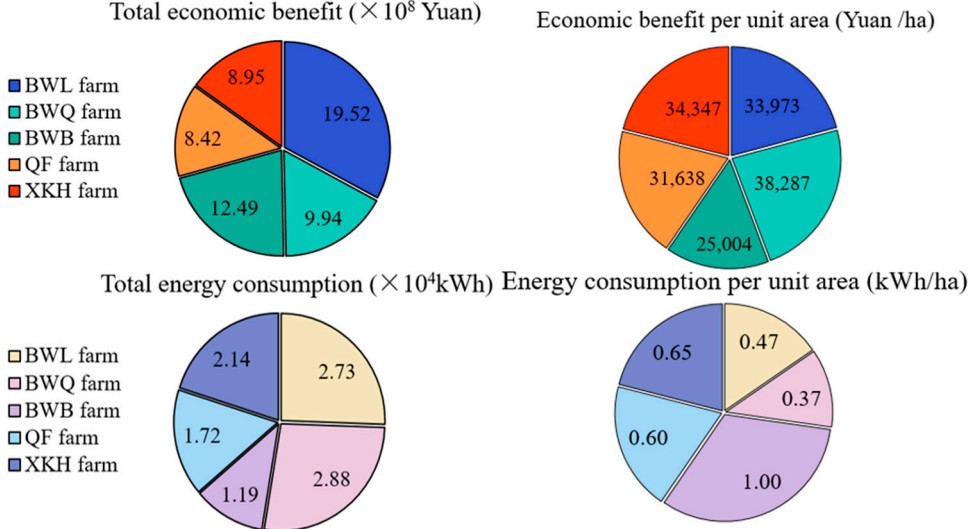

**Figure 8.** Individual farm energy target results.

*4.5. Analysis of Coordinated Development Degree under Future Change*

The research predicts the future coordinated development degree of agriculture from 2021 to 2030, as shown in Figure 9. The average coordinated development degree of the five farms from 2021 to 2025 is between 0.13 and 0.47. And the average annual growth rate is between −18.46% and 3.93%. The comparison shows that the coordinated development degree of the BWL farm continues to decline, while the coordinated development

degree of the BWQ farm shows a trend of first stability and then decline. Although the overall development of the BWQ farm is still the most balanced among the five farms, its downward trend needs to be vigilant. However, the BWB farm, the QF farm, and the XKH farm all show a trend of first increase and then decrease. Among them, the coordinated development degree of the BWB farm is the lowest, while the coordinated development degree of the XKH farm is the highest. On the whole, there are still great contradictions and problems in the balance between resources, economy, society, environment, and ecology in the five farms. Overall, significant contradictions and challenges persist in achieving a balanced equilibrium among resources, economy, society, environment, and ecology within the five major farms. This underscores the imperative for farm managers to actively promote coordinated development across multiple dimensions while concurrently reinforcing organic coordination among various facets, all while ensuring sustained harmony within their respective systems. Such efforts are essential to engendering a synergistic and interactive momentum, facilitating the sustainable and coordinated advancement of agriculture across multiple dimensions. In light of this, farm managers must steadfastly adhere to the principles of scientific development. They should prioritize and leverage the unique strengths of each farm within diverse leading industries and resource domains. Building upon this foundation, they can optimize operational layouts and continually refine industrial structures. Through these measures, the coordinated enhancement of economic, social, and environmental outcomes can be realized.

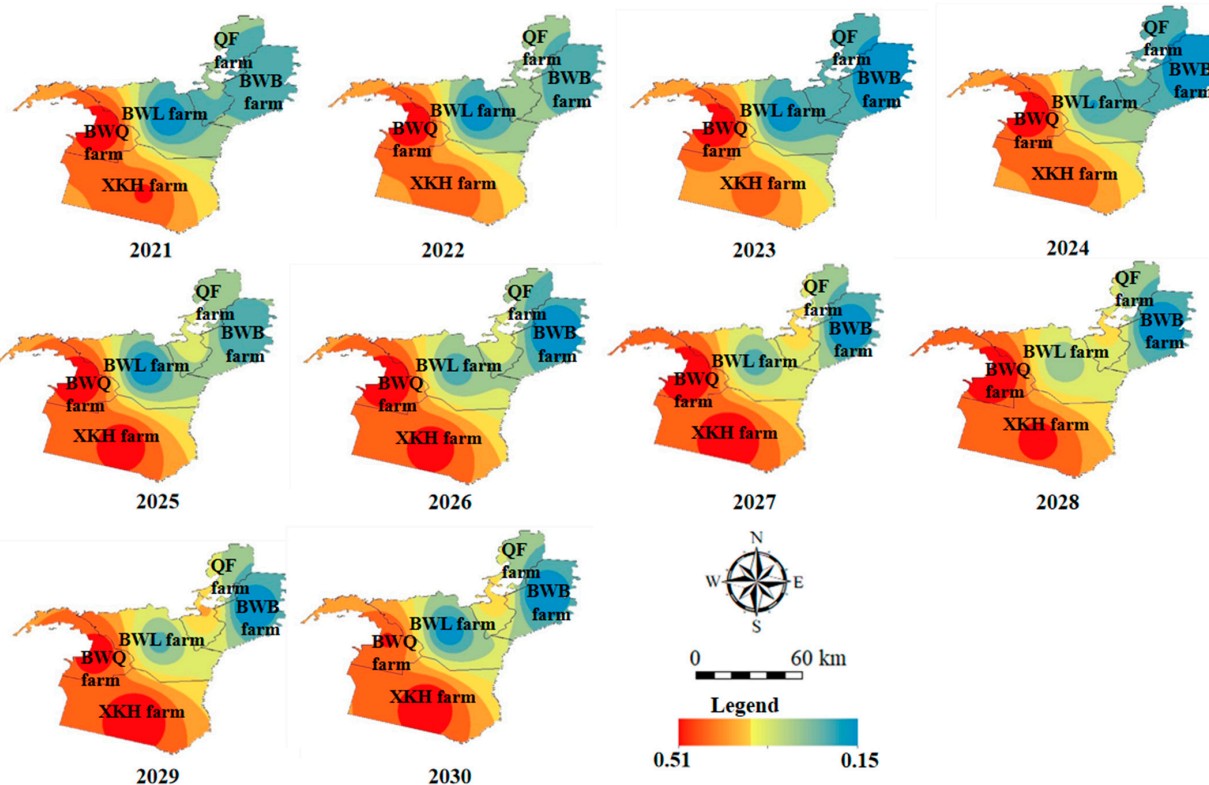

**Figure 9.** Calculation results of the coordinated development degree of each farm in 2021–2030.

### 4.6. Analysis of Water Rights Trading under Changing Environment

The changing environment in the future will have a significant impact on MCO-AWR-WPRCE. The SD model is used to simulate and forecast the evaluation indicators, and the coordinated development degree of each farm in 2021–2030 is obtained. The results are brought into MCO-AWR-WPRCE, and the results of water rights trading under future change scenarios are obtained, as shown in Figure 10. In 2021–2030, the BWL farm has been in a state of water shortage, and the situation continues to grow. The priority of purchasing water rights is as follows: the BWQ farm, then the XKH farm. In 2021, the initial water

rights of the farm can meet the farm demand, but in 2022–2030, the initial water rights allocation gradually fails to meet the farm's water demand, and it begins to purchase water rights from other farms. The priority levels of water rights purchased by the BWB farm are as follows: the QF farm, then the XKH farm. In the period of 2021–2030, the initial water rights allocation of the BWQ farm, the QF farm, and the XKH farm can always meet the needs of their own agricultural development, and the surplus water rights can be sold to obtain additional economic benefits.

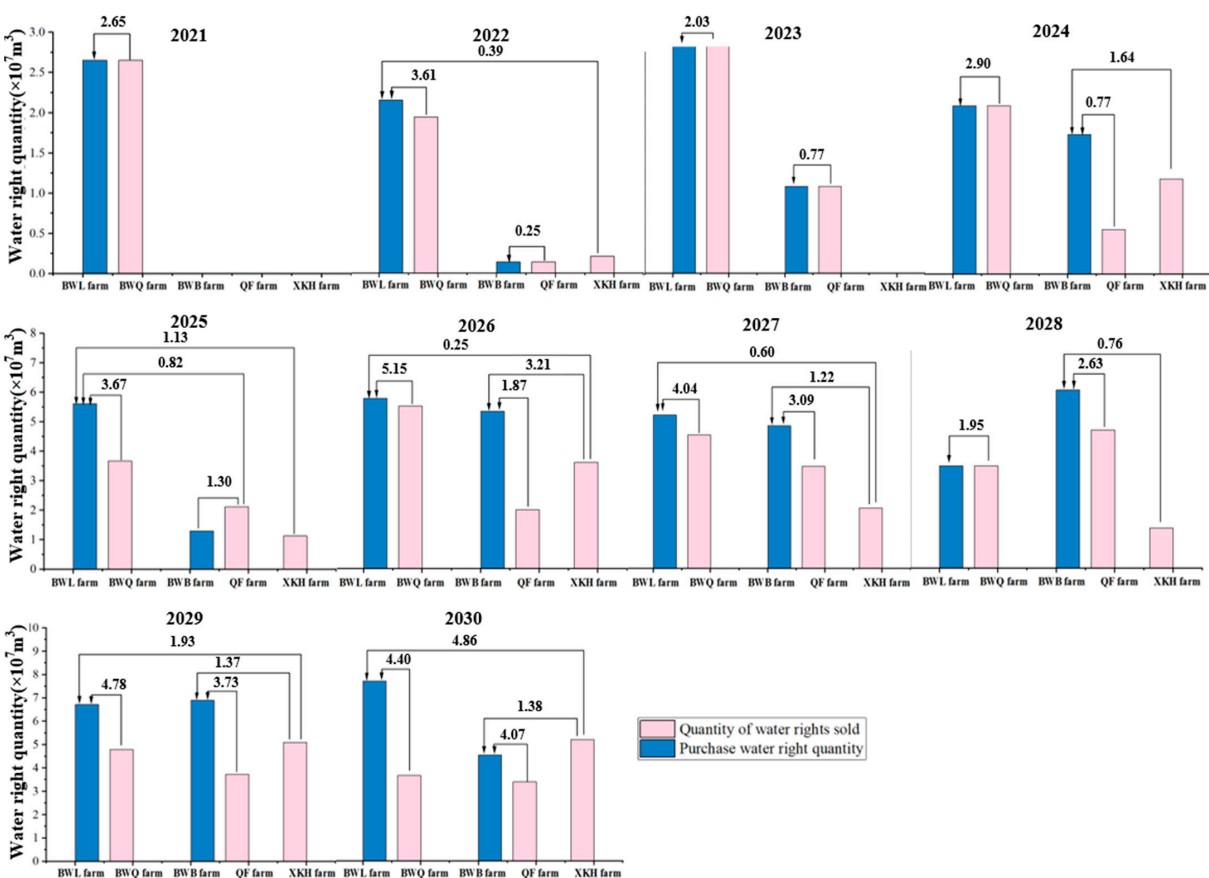

**Figure 10.** Results of water rights trading by farms in 2021–2030.

## 5. Discussion

At present, some studies try to take the local water shortage level as the basis to improve the water price model, and they are only limited to calculating the regional average water availability. These models are restricted by the current regional supervision water price paradigm [41,42]. In addition, in the current water price paradigm, all stations in a specific administrative unit will share the same charging standard. This price system violates the fair principle of sustainable development of water resources. Therefore, the whole water price model is adopted in the pricing process of water resources, and the trading distance of water rights in various regions is considered. It is reasonable to bring these factors into the pricing process, which can express the principle of fairness and rationality.

Among the parameters, the initial water right allocation weight exhibits the highest positive sensitivity to the optimization model (sensitivity = 4.31), indicating a substantial impact. Conversely, the water right transaction distance demonstrates the most prominent negative sensitivity to the optimization model (sensitivity = −1.63), in Table 9. While parameters of other coefficient types show correlations, their influence is comparatively weaker. To assess the overall sustainability of the model, considering the comprehensive development level of the economy, resource utilization efficiency, and energy consumption, we employed the harmony model. The harmony model calculated an initial harmony

degree of 0.32, reflecting the balance between economy, resource utilization efficiency, and energy consumption. Following optimization, the harmony degree of the model increased to 0.41, indicating enhanced overall sustainability and improved comprehensive benefits. These findings underscore the positive impact of model optimization on sustainability metrics, signifying an advancement in the balanced development of economy, resource utilization efficiency, and energy consumption.

**Table 9.** Sensitivity result.

| Variable | Sensitivity Result |
|---|---|
| Initial water right weight | 4.31 |
| Cost | −0.001 |
| Price | −0.01 |
| Water right transaction distance | −1.63 |
| Yield | 3.94 |
| Planting area | 2.85 |
| Water availability coefficient | 2.54 |

A key assumption of the model is that the transfer of water resources is affected by distance, and the energy consumption of water resources cannot be ignored in the process of transfer. In existing research, it is assumed that a virtual water bank will sell the surplus water resources from various areas to the water bank, and water-deficient areas will buy water rights from the water bank to achieve water savings. However, the water bank is merely a conceptual entity, providing only a theoretical basis for the process of water rights transactions. Additionally, there is no consideration of energy and resource consumption in the process of buying and selling water rights through the water bank, which does not align with reality. In this study, the channel is considered as the network for water rights trading, and the consumption of energy and resources in the process of water rights trading is taken into account, providing a robust theoretical foundation for the actual implementation of water rights trading schemes. Moreover, the theoretical model is analyzed using the actual irrigation area XHK, yielding positive results in terms of improved water resource utilization and economic benefits.

In this paper, the MCO-AWR-WPRCE model is proposed, and the model is studied on the scale of the irrigation area. Although this work is meaningful to all aspects of environmental management, some limitations need to be solved in the future. Secondly, the proposed MCO-AWR-WPRCE model has a strong connection with the current water price system, which is based on the overall rationality of the current water price system. The current water price system in the study area may have limitations. For example, the current system only considers the water price cost of agricultural water, which may be at a low level for the water price setting of industry and the service industry. In future studies, the researchers can adjust the model for specific regions and analyze the possibility of more water sources and treatment for the various processes of water intake, transport, use, and drainage.

## 6. Conclusions

In this study, an optimization model framework of agricultural water rights management based on a multi-dimensional evaluation system is constructed. The biggest highlight of this paper is to build a sound system of assessment and optimization of sustainable management of agricultural water rights under the environment of multiple uncertainties of the hydrological environment and management system and to weigh the economic benefits, energy consumption, and water use efficiency of agricultural water rights trading. The synergistic effects of multiple dimensions such as resources, economy, society, environment, and ecology on the initial allocation of water rights are quantified. The model is applied to the XKH irrigation area in Heilongjiang province, northeast China. The main conclusions are as follows:

(1) The use of water rights in the farms of BWL and BWB is in a relatively tight state, and water rights need to be purchased from other farms to meet their agricultural development needs. The transfer of water rights will tend to areas with higher comprehensive benefits and higher sustainable development level.

(2) The total economic benefit of each farm increased by 2.25% compared with the actual situation, and the water interest efficiency increased by 7.43%. The improved benefits indicate that MCO-AWR-WPRCE can improve agricultural water efficiency and increase economic benefits.

(3) In the case of future changes, the BWL farm has always been in a state of water shortage, and purchases water rights from the BWQ farm and the XKH farm. The initial water rights allocated by the BWB farm in 2021 can meet its own requirements, and then it needs to purchase water rights from the QF farm and the XKH farm to meet its own development needs.

**Author Contributions:** Conceptualization, L.S. and H.W.; methodology, L.S.; software, L.S.; validation, H.W.; formal analysis, L.D.; investigation, H.W.; resources, L.S.; data curation, L.S.; writing—original draft preparation, L.S.; writing—review and editing, H.W.; visualization, L.S.; supervision, H.W.; project administration, H.W.; funding acquisition, L.D. All authors have read and agreed to the published version of the manuscript.

**Funding:** This research was funded by Heilongjiang Province Postdoctoral Research Launch Fund, grant number LHB-Q16008.

**Data Availability Statement:** The original contributions presented in this study are included in the article; further inquiries can be directed to the corresponding author.

**Conflicts of Interest:** The authors declare no conflicts of interest.

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
