# Peer review of "Multi-Dimensional Collaborative Optimization Model for Agricultural Water Rights Based on Water Price Reform under Changing Environment"

_water, doi:10.3390/w16091262_

Round 1

Reviewer 1 Report

Comments and Suggestions for Authors

After reviewing the manuscript titled "Multi-dimensional collaborative optimization model for agricultural water rights based on water price reform under changing environment" by Song Linlin, Wang Hongshu, and Ding Liang, here are some questions and suggestions that could help clarify and potentially improve the paper:

Addressing these questions and suggestions could provide additional clarity, depth, and applicability to your work, making it a more valuable resource for researchers, policymakers, and practitioners in the field of water resources management and agricultural sustainability.

Could you provide a simplified overview or a conceptual diagram of the model to help readers understand its structure and components more intuitively?

It would be beneficial to discuss the key assumptions behind your model in more detail. How do these assumptions affect the model's applicability to real-world scenarios?

How was the model validated? Are there specific case studies or datasets that were used to test the accuracy and reliability of the model predictions?

A sensitivity analysis of the model’s key parameters could strengthen the paper. Which parameters have the most significant impact on the model outcomes, and how sensitive is the model to changes in these parameters?

The manuscript claims improvements in economic benefits and water resources utilization efficiency. Can you quantify these benefits and discuss any potential trade-offs or negative impacts? Could you discuss how social equity and environmental sustainability are addressed within the model? For instance, how does the model ensure fair water allocation among different stakeholders?

The model accounts for changing environmental conditions. Can you elaborate on how it adapts to long-term changes, such as climate change, and short-term variations, such as annual precipitation fluctuations?

Additionally, the choice of references should be supplemented with different models incorporating various factors. For example, consider adding references that explore diverse environmental studies and selected mathematical functions used for operation data information, such as: "Water, Resources, and Resilience: Insights from Diverse Environmental Studies." Water 2023, 15, 3965. https://doi.org/10.3390/w15223965. Valis, D.; Zak, L.; Walek, A. "Selected mathematical functions used for operation data information." In Safety, Reliability and Risk Analysis: Beyond the Horizon; CRC Press-Taylor & Francis Group: Boca Raton, FL, USA, 2014; pp. 1303–1308.

These references could provide a broader context and support for your model's design and application, enhancing the manuscript's depth and relevance to the field.It would be interesting to see a discussion on the model’s scalability and adaptability to different regions with varying water availability, agricultural practices, and economic conditions.

The manuscript could be strengthened by a clearer articulation of policy implications based on the model findings. What specific policy recommendations emerge, especially concerning water price reform? How can policymakers or water resource managers use your model in practice? Are there tools or interfaces planned to facilitate its use.

Author Response

Comment 1: Could you provide a simplified overview or a conceptual diagram of the model to help readers understand its structure and components more intuitively?

Response: We are grateful for this comment. According to your comments and suggestions, we added the concept map of the research model at the end of chapter 2.3, as shown below.

Figure 2. Model concept map”

Comment 2: It would be beneficial to discuss the key assumptions behind your model in more detail. How do these assumptions affect the model's applicability to real-world scenarios?

Response: We appreciate this comment. According to your comments and suggestions, we discussed the key assumptions of the study in more detail and analyzed the applicability of the assumptions to the model and the real scene, and added the supplementary contents to Chapter 5, as follows. “A key assumption of the model is that the transfer of water resources is affected by distance, and the energy consumption of water resources cannot be ignored in the process of transfer. In existing research, it is assumed that a virtual water bank will sell the surplus water resources from various areas to the water bank, and waterdeficient areas will buy water rights from the water bank to achieve water savings. However, the water bank is merely a conceptual entity, providing only a theoretical basis for the process of water rights transactions. Additionally, there is no consideration of energy and resource consumption in the process of buying and selling water rights through the water bank, which does not align with reality. In this study, the channel is considered as the network for water rights trading, and the consumption of energy and resources in the process of water rights trading is taken into account, providing a robust theoretical foundation for the actual implementation of water rights trading schemes. Moreover, the theoretical model is analyzed using the actual irrigation area XHK, yielding positive results in terms of improved water resource utilization and economic benefits.” 

Comment 3: How was the model validated? Are there specific case studies or datasets that were used to test the accuracy and reliability of the model predictions?

Response: We appreciate this comment. To assess the validity of the system dynamics model, we conducted a series of tests. The model test aimed to measure the disparity between historical data and simulation data, thereby evaluating their similarity and determining the reliability and accuracy of the model's simulation. Subsequently, after the system dynamics model test, it was imperative to evaluate the model itself to ensure the accuracy of its parameter calculations. We utilized the root mean square error (RMSE) method for model evaluation. A lower RMSE value indicates higher model credibility and superior performance. This evaluation method was applied to Chapter 3.2.2, with the following details outlined.

 “After establishing the system dynamics model, it's crucial to assess its validity. This entails testing the model's accuracy by comparing historical data with simulation results to determine their consistency and gauge the reliability of the model. Subsequent to the system dynamics model test, evaluating the model is essential to ensure the precision of its parameter calculations. The root mean square error (RMSE) method is employed for this evaluation. The results of RMSE are shown in Figure 5.” 

Comment 4: A sensitivity analysis of the model’s key parameters could strengthen the paper. Which parameters have the most significant impact on the model outcomes, and how sensitive is the model to changes in these parameters?

Response: Thank you. According to your comments and suggestions, we conduct sensitivity analysis on the model results, and add the results to Chapter 5. The details are as follows.

 “To investigate the impact of various factors on the model's objectives, we conducted sensitivity analysis for each parameter. A sensitivity threshold of ∣ sensitivity ∣ ≥ 0.1 was employed, where values exceeding this threshold indicate significant influence on the system and close correlation, while values below it suggest minor influence and relative insensitivity. Furthermore, sensitivity analysis reveals both positive and negative sensitivities. A positive value signifies that the variable and parameter change in the same direction, meaning an increase or decrease in the parameter results in a corresponding increase or decrease in the variable. Conversely, a negative value indicates an inverse relationship between the variable and parameter. Among the parameters, the initial water right allocation weight exhibits the highest positive sensitivity to the optimization model (sensitivity = 4.31), indicating a substantial impact. Conversely, the water right transaction distance demonstrates the most prominent negative sensitivity to the optimization model (sensitivity = -1.63). While parameters of other coefficient types show correlations, their influence is comparatively weaker…”

Comment 5: The manuscript claims improvements in economic benefits and water resources utilization efficiency. Can you quantify these benefits and discuss any potential trade-offs or negative impacts? Could you discuss how social equity and environmental sustainability are addressed within the model? For instance, how does the model ensure fair water allocation among different stakeholders?

Response: Thank you for your comment. The manuscript highlights improvements in economic benefits and water resource utilization efficiency, albeit with a noted negative impact on energy consumption. To assess the overall sustainability of the model, considering the comprehensive development level of the economy, resource utilization efficiency, and energy consumption, we employed the harmony model. The harmony model calculated an initial harmony degree of 0.32, reflecting the balance between economy, resource utilization efficiency, and energy consumption. Following optimization, the harmony degree of the model increased to 0.41, indicating enhanced overall sustainability and improved comprehensive benefits. These findings underscore the positive impact of model optimization on sustainability metrics, signifying an advancement in the balanced development of economy, resource utilization efficiency, and energy consumption. The following content has been added to Chapter 5 to elaborate on these insights.

“…To assess the overall sustainability of the model, considering the comprehensive development level of the economy, resource utilization efficiency, and energy consumption, we employed the harmony model. The harmony model calculated an initial harmony degree of 0.32, reflecting the balance between economy, resource utilization efficiency, and energy consumption. Following optimization, the harmony degree of the model increased to 0.41, indicating enhanced overall sustainability and improved comprehensive benefits. These findings underscore the positive impact of model optimization on sustainability metrics, signifying an advancement in the balanced development of economy, resource utilization efficiency, and energy consumption.”

Comment 6: The model accounts for changing environmental conditions. Can you elaborate on how it adapts to long-term changes, such as climate change, and short-term variations, such as annual precipitation fluctuations?

Response: Thank you for your comments. The changing environmental conditions depicted in the model correspond to future prediction scenarios of model parameters derived from the system dynamics model. Consequently, the final water right allocation scheme is informed by forecasting future changes in these parameters. Regrettably, our study did not incorporate considerations of climate change and precipitation alterations. This omission stemmed from the perceived lack of a clear relationship between the primary focus of the model (multi-objective optimization process) and climate change dynamics. However, we acknowledge the importance of this aspect and recognize the necessity of addressing it in future research endeavors. Moving forward, we intend to delve deeper into the implications of climate change on water rights allocation, incorporating your valuable insights and recommendations into our future studies.

Comment 7: Additionally, the choice of references should be supplemented with different models incorporating various factors. For example, consider adding references that explore diverse environmental studies and selected mathematical functions used for operation data information, such as: "Water, Resources, and Resilience: Insights from Diverse Environmental Studies." Water 2023, 15, 3965. https://doi.org/10.3390/w15223965. Valis, D.; Zak, L.; Walek, A. "Selected mathematical functions used for operation data information." In Safety, Reliability and Risk Analysis: Beyond the Horizon; CRC Press-Taylor & Francis Group: Boca Raton, FL, USA, 2014; pp. 1303–1308.

Response: Thank you for your comments. According to your opinion, we added relevant references.

Comment 8: These references could provide a broader context and support for your model's design and application, enhancing the manuscript's depth and relevance to the field. It would be interesting to see a discussion on the model’s scalability and adaptability to different regions with varying water availability, agricultural practices, and economic conditions. The manuscript could be strengthened by a clearer articulation of policy implications based on the model findings. What specific policy recommendations emerge, especially concerning water price reform? How can policymakers or water resource managers use your model in practice? Are there tools or interfaces planned to facilitate its use.

Response: Thank you for your comments. In conclusion, our study suggests an increase in the price of water resources as a means to curtail wastage and foster a more sustainable development of the water rights trading market. Decisionmakers are encouraged to utilize the model in their respective research domains by substituting the input data with actual data pertinent to their regions, thereby deriving customized water rights trading schemes. It is important to note that while the model has been developed to a programming stage, it currently lacks a specific tool or interface for user interaction. Moving forward, our research will focus on further refining the model for practical application and implementation. By advancing the development and application of this model, we aim to contribute to the efficient and equitable allocation of water resources, thereby promoting sustainable water management practices.

Reviewer 2 Report

Comments and Suggestions for Authors

Identifying the key elements of agricultural water rights trading and their interactions is important for solving agricultural water scarcity. This manuscript provides an in-depth discussion of the importance and management of agricultural water rights, as well as proposing a multi-dimensional collaborative optimization model of agricultural water rights based on water pricing reform. The manuscript is well organized and contributes to improve the overall coordinated development of farms. I suggest a minor revision. my Specific comments are as follows:

1) In line 495,“XKH irrigation area” should be given a full name when it first appears.

2) In line 499, “BWL farm, BWQ farm, BWB farm, QF farm and XKH farm” should be given a full name when it first appears.

3) In figure 3, please add a description of what the different colors refer to.

4) In lines 576-581, change frequency is odd here, please consider restructure it. Additionally, the sentence structure can be improved for better flow.

5) In section "Analysis of coordinated development degree under future change", please briefly describe what the impacts of different degrees of coordinated development under future change would be on the farm, and any suggestions or ways to increase or decrease the degree of coordinated development on the farm.

6) In figure 9, suggest that each row of small pictures share a common vertical coordinate title, or all small pictures share a common vertical coordinate title; all small pictures share a common label, to make the picture more concise and intuitive.

7) Some long sentences are present in this article. Please break down the long sentences into shorter, more understandable sentences to improve readability and comprehension.

8) Some expressions and grammatical errors appear in this paper, please check and revise.

Comments on the Quality of English Language

 Moderate editing of English language required

Author Response

Comment 1: In line 495,“XKH irrigation area should be given a full name when it first appears.

Response: We are grateful for this comment. I'm very sorry that the full name of XKH irrigation area was not given, because the relevant information of XKH irrigation area is classified in China, so the research is replaced by abbreviation.

Comment 2: In line 499, “BWL farm, BWQ farm, BWB farm, QF farm and XKH farm” should be given a full name when it first appears.

Response: We appreciate this comment. I'm very sorry that I didn't give the full name of the farm in XKH irrigation area. This is because the relevant information of XKH irrigation area is classified in China, including the farm in the irrigation area, so the name of the farm in the irrigation area is also replaced by abbreviation. 

Comment 3: In figure 3, please add a description of what the different colors refer to.

Response: We appreciate this comment. According to your comments and suggestions, we add notes at the bottom of Figure 4 (original Figure 3) to explain the meaning of each color, as follows.

Figure 4. Multi-dimensional evaluation of SD flow charts

Note: Blue represents the environmental dimension, red represents the resource dimension, pink represents the economic dimension, green represents the social dimension and yellow represents the ecological dimension.”

Comment 4: In lines 576-581, “change frequency” is odd here, please consider restructure it. Additionally, the sentence structure can be improved for better flow.

Response: Thank you. According to your comments and suggestions, we changed "changed frequency" into "changed relatively" to make the meaning of the sentence more accurate. The details are as follows.

 “The error of the simulation results is controlled within 10%, which verifies the reliability of the model. In the resource dimension, key indicators such as actual evapotranspiration, precipitation and fertilizer application are selected for simulation prediction. Compared with the historical values, it can be seen that the actual evapotranspiration, precipitation and fertilizer application showed a gradual upward trend, among which the change trend of actual evapotranspiration is relatively gentle, and the changed relatively between the historical minimum actual evapotranspiration and the predicted maximum actual evapotranspiration is 56.81%.”

Comment 5: In section "Analysis of coordinated development degree under future change", please briefly describe what the impacts of different degrees of coordinated development under future change would be on the farm, and any suggestions or ways to increase or decrease the degree of coordinated development on the farm.

Response: Thank you for your comment. According to your opinions and suggestions, we have added in Chapter 4.5 the influence of different coordinated development degrees on the farm under future changes, and the suggestions to improve or reduce the coordinated development degree of the farm, as follows.

“…Overall, significant contradictions and challenges persist in achieving a balanced equilibrium among resources, economy, society, environment, and ecology within the five major farms. This underscores the imperative for farm managers to actively promote coordinated development across multiple dimensions while concurrently reinforcing organic coordination among various facets, all while ensuring sustained harmony within their respective systems. Such efforts are essential to engendering a synergistic and interactive momentum, facilitating the sustainable and coordinated advancement of agriculture across multiple dimensions. In light of this, farm managers must steadfastly adhere to the principles of scientific development. They should prioritize and leverage the unique strengths of each farm within diverse leading industries and resource domains. Building upon this foundation, they can optimize operational layouts and continually refine industrial structures. Through these measures, the coordinated enhancement of economic, social, and environmental outcomes can be realized.”

Comment 6: In figure 9, suggest that each row of small pictures share a common vertical coordinate title, or all small pictures share a common vertical coordinate title; all small pictures share a common label, to make the picture more concise and intuitive.

Response: Thank you for your comments. According to your opinion, we will modify Figure 10 (original Figure 9) as follows.

Figure 10. Results of water rights trading by farms in 2021-2030”

Comment 7: Some long sentences are present in this article. Please break down the long sentences into shorter, more understandable sentences to improve readability and comprehension.

Response: Thank you for your comments. According to your comments and suggestions, we have revised the long sentences in the full text and ensured that there are no grammatical and word errors.

Comment 8: Some expressions and grammatical errors appear in this paper, please check and revise.

Response: Thank you for your comments. According to your comments and suggestions, we checked and read the full text, and made sure there were no grammatical and word errors.

Reviewer 3 Report

Comments and Suggestions for Authors

This paper discusses an important issue which is also crosscutting, bringing sustainability issues, water, and food together. In fact, this paper discusses that agricultural water rights trading is increasingly seen as a solution to address water shortages in agriculture. However, the complexities of this trading system and its interactions are not fully understood, especially in uncertain environments. Thus, this study develops a comprehensive optimization model for agricultural water rights, focusing on water price reforms amid changing conditions. The model evaluates the combined impacts of various objectives—resource, economic, social, environmental, and ecological—on initial water rights allocation and trading. Additionally, it incorporates system dynamics and intuitionistic fuzzy numbers to account for environmental changes and management uncertainties. Results demonstrate that water rights trading enhances overall farm development, improving economic benefits and water resource efficiency.

The paper is well structured and justified, but it would benefit from revisions. In particular, my suggestions are as follows:

- please clearly state the research question at the top of the introduction

- state what is the literature you are contributing to; what kind of contribution is it (empirical or theoretical?)?

- discuss also water rights in terms of water reallocation practices; see for instance the work of Timothy Liptrot published in Water Alternatives on the case of Jordan

- What about agroecological practices and the role of centralised vs decentralised practices? See the work of Giuliano Martiniello on this, and also of Ariane Goetz et al on the case of the MENA region's poly centric governance

- Would the concept of virtual water also be relevant? or could you contextualise this study also in transboundary water issues and dynamics?

Happy to read an updated version.

Author Response

Comment 1: please clearly state the research question at the top of the introduction

Response: We are grateful for this comment. According to your opinion, we clearly stated the research questions in the first paragraph of the introduction, and the specific contents are as follows.

“This study proposed a multi-dimensional collaborative optimization model for agricultural water rights based on water price reform under changing environment(MCO-AWR-WPRCE)to weigh the contradictions among multiple objectives of economic benefits, energy consumption and water use efficiency of agricultural water rights trading. MCO-AWR-WPRCE can effectively manage water rights in irrigated areas to reduce agricultural water use and enhance agricultural benefit value while ensuring agricultural production, providing valuable insights and guidance for current and future water resources management.”

Comment 2: In line 499, “BWL farm, BWQ farm, BWB farm, QF farm and XKH farm” should be given a full name when it first appears.

Response: We appreciate this comment. According to your opinion, in the last paragraph of the introduction, we explained the nature of the research content and its contribution to production and life. The specific contents are as follows. “MCO-AWR-WPRCE model quantifies the synergistic effects of resources, economy, society, environment and ecology on the initial water rights allocation, and considers the impacts of multiple uncertainties of changing environment and management system on the model, so as to obtain a reasonable agricultural water rights allocation scheme. MCO-AWR-WPRCE model is a theoretical model of water rights trading considering initial water rights allocation and water price reform. By managing water rights trading schemes, agricultural water consumption is reduced, thus achieving the purpose of water saving. And ensure the agricultural production demand in water-deficient areas.” 

Comment 3: discuss also water rights in terms of water reallocation practices; see for instance the work of Timothy Liptrot published in Water Alternatives on the case of Jordan

Response: We appreciate this comment. According to your comments and suggestions, we have studied the research on Jordan published by Timothy Liptrot in Substitute for Water, and made a brief analysis of the redistribution of water resources, which is added to the introduction. The details are as follows.

“Water right transaction is actually the redistribution of water resources. The redistribution of water resources can improve economic benefits, but some people are conservative because they exaggerate their economic benefits, because the increase of economic benefits will do harm to other aspects (such as environment and resource consumption). According to Liptrot and Hussein, redistribution may reduce farmers' income and increase food prices[43]. In addition, the affected farmers themselves may not benefit from the growth of the waterreceiving sector. Therefore, the transaction price and reasonable formulation of water rights are particularly important. The trading price of water rights can solve the contradiction between the static nature of initial water rights allocation and the dynamics of social and economic development [14], and play a key role in the smooth realization of water rights trading. Too low trading price not only damages the interests of the transferor of water rights, but also is detrimental to the conservation and protection of water resources. The excessively high trading price may increase the water cost of the water rights transferor and make the transaction difficult to realize [5]. Therefore, determining the trading price of water rights is not only conducive to creating an effective water rights market, but also beneficial for solving the contradiction of spatial and temporal distribution of water resources and promoting the optimal allocation of water resources.”

Comment 4: What about agroecological practices and the role of centralised vs decentralised practices? See the work of Giuliano Martiniello on this, and also of Ariane Goetz et al on the case of the MENA region's poly centric governance

Response: Thank you. This study is aimed at the distribution and trading of agricultural water resources, which is not closely related to agricultural ecological practice and centralized and decentralized practice. However, we have thoroughly studied the research content and direction recommended by you, and the next research direction will be discussed and studied in depth according to your opinions and suggestions.

Comment 5: Would the concept of virtual water also be relevant? or could you contextualise this study also in transboundary water issues and dynamics?

Response: Thank you for your comment. This article is about the study of water right transaction allocation scheme, which is a real study of water transfer. Virtual water is the amount of water resources needed in agricultural production, that is, the virtual water condensed in products and services, so the correlation between them is not strong. Transboundary water is a water resource that spans different regions. The source of water resources studied is the allocation of water resources by managers, which belongs to the same region. However, virtual water and transboundary water are very interesting definitions. In the next research, we will conduct indepth research according to the direction you mentioned and combine it with the research content.

Round 2

Reviewer 1 Report

Comments and Suggestions for Authors

Please make bigger font of the following figure: Figure 9. Calculation results of the coordinated development degree of each farm in 729 2021-2030  and Figure 10. Results of water rights trading by farms in 2021-2030 and Figure 4. Multi-dimensional evaluation of SD flow charts.

Author Response

Comment 1: Please make bigger font of the following figure: Figure 9. Calculation results of the coordinated development degree of each farm in 729 2021-2030 and Figure 10. Results of water rights trading by farms in 2021-2030 and Figure 4. Multi-dimensional evaluation of SD flow charts.

Response: We are grateful for this comment. According to your opinion, we have enlarged the text in Figure 9, Figure 10 and Figure 4, as follows.

Figure 10. Results of water rights trading by farms in 2021-2030”

Figure 9. Calculation results of the coordinated development degree of each farm in 2021-2030”

Figure 4. Multi-dimensional evaluation of SD flow charts

Reviewer 3 Report

Comments and Suggestions for Authors

Looks better 

Author Response

Thank you very much for your efforts.